# HPV upregulates MARCHF8 ubiquitin ligase and inhibits apoptosis by degrading the death receptors in head and neck cancer

**Mohamed I. Khalil[1,2], Canchai Yang[1], Lexi Vu[1], Smriti Chadha[1], Harrison Nabors[1], Craig Welbon[3], Claire D. James[4], Iain M. Morgan[4], William C. Spanos[3], Dohun Pyeon[1]***

**1** Department of Microbiology and Molecular Genetics, Michigan State University, East Lansing, Michigan, United States of America, **2** Department of Molecular Biology, National Research Centre, El-Buhouth St., Cairo, Egypt, **3** Cancer Biology and Immunotherapies Group, Sanford Research, Sioux Falls, South Dakota, United States of America, **4** Philips Institute for Oral Health Research, School of Dentistry, Virginia Commonwealth University, Richmond, Virginia, United States of America

* dpyeon@msu.edu

**Data Availability Statement:** The raw data have been uploaded to Dryad: Khalil, Mohamed et al. (2023), HPV upregulates MARCHF8 ubiquitin ligase and inhibits apoptosis by degrading the

## Abstract

The membrane-associated RING-CH-type finger ubiquitin ligase MARCHF8 is a human homolog of the viral ubiquitin ligases Kaposi's sarcoma herpesvirus K3 and K5 that promote host immune evasion. Previous studies have shown that MARCHF8 ubiquitinates several immune receptors, such as the major histocompatibility complex II and CD86. While human papillomavirus (HPV) does not encode any ubiquitin ligase, the viral oncoproteins E6 and E7 are known to regulate host ubiquitin ligases. Here, we report that MARCHF8 expression is upregulated in HPV-positive head and neck cancer (HNC) patients but not in HPV-negative HNC patients compared to normal individuals. The *MARCHF8* promoter is highly activated by HPV oncoprotein E6-induced MYC/MAX transcriptional activation. The knockdown of MARCHF8 expression in human HPV-positive HNC cells restores cell surface expression of the tumor necrosis factor receptor superfamily (TNFRSF) death receptors, FAS, TRAIL-R1, and TRAIL-R2, and enhances apoptosis. MARCHF8 protein directly interacts with and ubiquitinates the TNFRSF death receptors. Further, MARCHF8 knockout in mouse oral cancer cells expressing HPV16 E6 and E7 augments cancer cell apoptosis and suppresses tumor growth in vivo. Our findings suggest that HPV inhibits host cell apoptosis by upregulating MARCHF8 and degrading TNFRSF death receptors in HPV-positive HNC cells.

## Author summary

Since host cell survival is essential for viruses to replicate persistently, many viruses have evolved to prevent host cell apoptosis. The human papillomavirus (HPV) oncoproteins are known to dysregulate proapoptotic proteins. However, our understanding of detailed mechanisms for HPV to inhibit apoptosis is limited. Here, we report that HPV induces expression of the membrane-associated ubiquitin ligase MARCHF8, which is upregulated in HPV-positive head and neck cancer. MARCHF8 ubiquitinates the tumor necrosis factor receptor superfamily (TNFRSF) death receptors, FAS, TRAIL-R1, and TRAIL-R2 for degradation. We further revealed that downregulation of the death receptors by

death receptors in head and neck cancer, Dryad, Dataset, https://doi.org/10.5061/dryad.ffbg79d04. All other relevant data are within the manuscript and its Supporting Information files.

**Funding:** This work was supported by funding from National Institutes of Health (R01 DE029524 and R01 DE026125 to W.C.S and D.P). The funders played no role in study design, data collection and analysis, decision to publish, or preparation of the manuscript.

**Competing interests:** The authors have declared that no competing interests exist.

MARCHF8 prevents cancer cell apoptosis and that knockout of MARCHF8 expression significantly inhibits in vivo tumor growth and enhances tumor-free survival of mice transplanted with mouse oral cancer cells expressing HPV16 E6 and E7. These results suggest that virus-induced degradation of death receptors leads to cancer cell survival in HPV-positive head and neck cancer.

## Introduction

High-risk human papillomavirus (HPV) is associated with about 5% of all human cancers, including ~25% of head and neck cancers (HNCs) [1–3]. HPV-positive HNC (HPV+ HNC) arises mainly in the oropharynx, while HPV-negative HNC (HPV- HNC), linked to smoking and drinking, develops in various mouth and throat regions [4,5]. The HPV+ HNC incidence has increased dramatically in recent decades [6–8]. The HPV oncoproteins E6 and E7 contribute to fundamental oncogenic mechanisms in HPV+ HNC development that require multiple oncogenic driver mutations in HPV- HNC [9,10].

Virus-induced ubiquitination and degradation of death receptors is a potent mechanism for host cell survival and viral replication [11]. Several DNA viruses downregulate the surface expression of the tumor necrosis factor receptor superfamily (TNFRSF) death receptors such as FAS and tumor necrosis factor (TNF)-related apoptosis-inducing ligand receptors (TRAIL-R1 and TRAIL-R2). For example, adenoviral E3 and human cytomegalovirus UL141 proteins inhibit apoptosis by inducing internalization and lysosomal degradation of the TNFRSF death receptors [12–14]. In addition, HPV16 E6 protein binds to the FAS-associated death domain and prevents FAS-induced apoptosis [15].

The membrane-associated RING-CH-type finger (MARCHF) proteins are a subfamily of the RING-type E3 ubiquitin ligase family [16]. The MARCHF proteins contain a C4HC3-type RING domain, initially identified in K3 and K5 ubiquitin ligases of Kaposi's sarcoma-associated herpesvirus (KSHV) [17–19]. MARCHF8, a MARCHF family member, is expressed in various tissue types and localizes in the plasma and endosome membranes [20]. Like KSHV K3 and K5 proteins, MARCHF8 ubiquitinates immunoreceptors such as the major histocompatibility complex II (MHC-II) [21] and CD86 [22], the cell adhesion molecules CD98 and CD44 [23], and the death receptor TRAIL-R1 (a.k.a., death receptor 4 and TNFRSF10A) [24]. Previous studies have shown that *MARCHF8* expression is upregulated in esophageal, colorectal, and gastric cancers [25–27]. However, the role of MARCHF8 in HPV-associated cancers is largely elusive despite its importance in regulating immune and death receptors.

Here, we report that HPV induces the viral escape of host cell apoptosis through upregulation of *MARCHF8* expression. HPV16 E6 activates *MARCHF8* promoter activity through MYC/MAX activation. Increased MARCHF8 protein in HPV+ HNC cells binds to and ubiquitinates FAS, TRAIL-R1, and TRAIL-R2 proteins. Knockdown and knockout of *MARCHF8* expression in human and mouse HPV+ HNC cells significantly enhances apoptosis and attenuates in vivo tumor growth. Our findings provide a novel insight into virus-induced evasion of host cell apoptosis and a potential therapeutic target to treat HPV+ HNC.

## Results

### MARCHF8 expression is significantly upregulated in HPV+ HNC by HPV oncoproteins

To determine if *MARCHF8* expression levels are altered in HPV+ and HPV- HNC patients, we analyzed our gene expression data from HPV+ (*n* = 16) and HPV- (*n* = 26) HNC patients

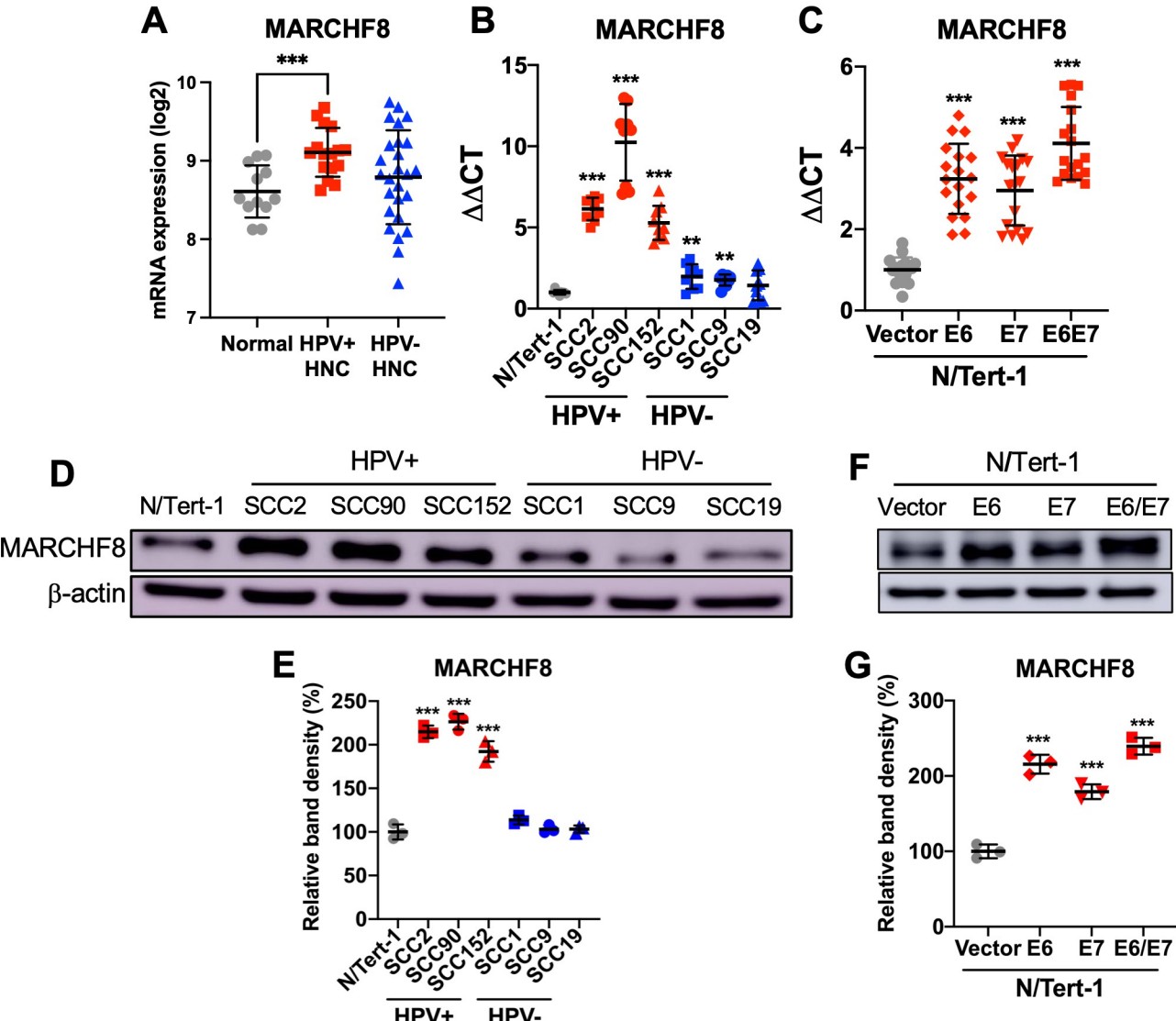

**Fig 1. *MARCHF8* expression is upregulated in HPV+ HNC.** MARCHF8 mRNA expression levels in microdissected human tissue samples from HPV
+ (*n* = 16) and HPV- (*n* = 26) HNC patients, and normal individuals (*n* = 12) were analyzed using our previous gene expression data (GSE6791) [28]
and shown as fluorescence intensity (log2) (**A**). The MARCHF8 mRNA expression was validated in normal (N/Tert-1), HPV+ HNC (SCC2, SCC90, and
SCC152), and HPV- HNC (SCC1, SCC9, and SCC19) cells (**B**) by RT-qPCR. MARCHF8 mRNA levels were determined in N/Tert-1 cells expressing E6
and/or E7 by RT-qPCR (**C**). The data shown are ΔΔCT values normalized by the GAPDH mRNA level as an internal control. MARCHF8 protein levels
were determined in HPV+ and HPV- HNC cells (**D and E**) and N/Tert-1 cells expressing HPV16 E6, E7, or E6 and E7 (**F and G**) by western blotting. β-
actin was used as a loading control. The relative band density was quantified using the NIH ImageJ program (**E and G**). All experiments were repeated
at least three times, and the data shown are means ± SD. *P* values were determined by Student's *t*-test. $^*p < 0.05$, $^{**}p < 0.01$, $^{***}p < 0.001$.

and normal individuals (*n* = 12) [28]. Our result showed that MARCHF8 mRNA expression in
HPV+ HNC patients, but not in HPV- HNC patients, is significantly upregulated compared to
the normal individuals (**Fig 1A**). Although several HPV- HNC patients also showed high
MARCHF8 mRNA levels, there were no statistically significant differences compared to nor-
mal individuals and HPV+ HNC patients (**Fig 1A**). We also measured MARCHF8 mRNA lev-
els in HPV+ HNC cell lines (SCC2, SCC90, and SCC152), HPV- HNC cell lines (SCC1, SCC9,
and SCC19), and normal hTERT immortalized keratinocytes (N/Tert-1) by RT-qPCR. Our
results showed significantly higher levels of MARCHF8 mRNA in all HPV+ HNC cells

compared to N/Tert-1 and HPV- HNC cells (**Fig 1B**). Next, to assess if HPV16 E6 and/or E7 are sufficient to upregulate *MARCHF8* expression, we established in N/Tert-1 cells expressing HPV16 E6 and/or E7 using lentiviral transduction and puromycin selection. The HPV16 E6 and E7 mRNA expression and E7 protein levels were determined in these N/Tert-1 cells (**S1A–S1C Fig**). MARCHF8 mRNA levels were measured in N/Tert-1 vector, N/Tert-1 E6, N/Tert-1 E7, and N/Tert-1 E6E7 cells by RT-qPCR. The results showed that expression of either or both E6 and E7 was sufficient to significantly upregulate MARCHF8 mRNA expression in N/Tert-1 cells (**Fig 1C**). Western blotting analyses demonstrated that MARCHF8 protein levels are also significantly higher in HPV+ HNC (**Fig 1D and 1E**) and E6/E7 expressing N/Tert-1 cells (**Fig 1F and 1G**) compared to N/Tert-1. These results suggest that *MARCHF8* expression is upregulated in HPV+ HNC cells by the HPV oncoproteins E6 and E7. To determine the mechanism by which HPV16 E6 and E7 increase MARCHF8 expression, we measured MARCHF8 mRNA and protein levels in N/Tert-1 cells that express E6 8S9A10T (deficient in p53 binding) [29], E6 I128T (deficient in E6AP binding) [30], or E7 ΔDLYC (deletion of the pRb binding domain) [31,32]. The results showed that none of these E6 and E7 mutations abrogated E6/E7-mediated upregulation of MARCHF8 expression in N/Tert-1 cells (**S2A–S2C Fig**). These findings suggest that the upregulation of *MARCHF8* expression by HPV16 E6 and E7 in normal keratinocytes is independent of p53 and pRb degradation, respectively.

## The HPV oncoprotein E6 induces *MARCHF8* promoter activity through the MYC/MAX complex

To assess the transcriptional regulation of MARCHF8 mRNA expression by HPV oncoproteins, *MARCHF8* promoter activity was analyzed in HPV+ and HPV- HNC cells. We first cloned several different lengths (0.17 to 1.0 kb) of the *MARCHF8* promoter regions between -840 and +160 into a luciferase reporter plasmid, pGL4.2 (**Figs 2A and S3A**), and tested their transcriptional activity in HPV+ HNC (SCC152) and HPV- HNC (SCC1) cells. Our results showed that the promoter activities of all fragments are higher in HPV+ HNC cells compared to HPV- HNC cells (**Fig 2B**). Furthermore, the promoter region from -60 to -10 contains target sequences for cis-acting elements essential for *MARCHF8* promoter activity (**Figs 2C and S3A**). To further determine if HPV oncoprotein expression increases *MARCHF8* promoter activity in HPV- HNC cells, we cotransfected SCC1 cells with the *MARCHF8* promoter-reporter construct and HPV16 E6 and/or E7 expression plasmids. Interestingly, the expression of HPV16 E6 or E6E7, but not E7 alone, significantly increases *MARCHF8* promoter activity (**Fig 2D**). Given that both E6 and E7 expression upregulates MARCHF8 mRNA expression in N/Tert-1 cells (**Fig 1C**), these results suggest that while E6 directly activates the *MARCHF8* promoter, E7 increases the MARCHF8 mRNA level through an alternative mechanism.

To examine whether *MARCHF8* promoter activation by HPV16 E6 is mediated by p53 degradation, we measured MARCHF8 promoter activity in HPV- HNC cells (SCC1) expressing E6 Y54D (deficient in p53 binding) [33,34] and E6 I128T, using an extended MARCHF8 promoter region [1.5 kb] (**S3B Fig**). The results showed that both E6 Y54D and E6 I128T retained the E6 function of activating the *MARCHF8* promoter (**S3B Fig**), suggesting that p53 degradation is not required for *MARCHF8* promoter activation. As HPV16 E7 expression upregulates *MARCHF8* expression in N/Tert-1 cells (**Fig 1C and 1F**) but does not induce the activity of the essential *MARCHF8* promoter [0.25 kb] (**Fig 2D**), we tested if E7 affects the extended *MARCHF8* promoter region [1.5 kb]. The results showed that neither wildtype E7 nor E7 ΔDLYC induced promoter activity with up to the 1.5 kb upstream region (**S3B Fig**). These results suggest that HPV16 E7 upregulates *MARCHF8* expression through a different mechanism(s), which is independent from the immediate upstream promoter region of *MARCHF8*.

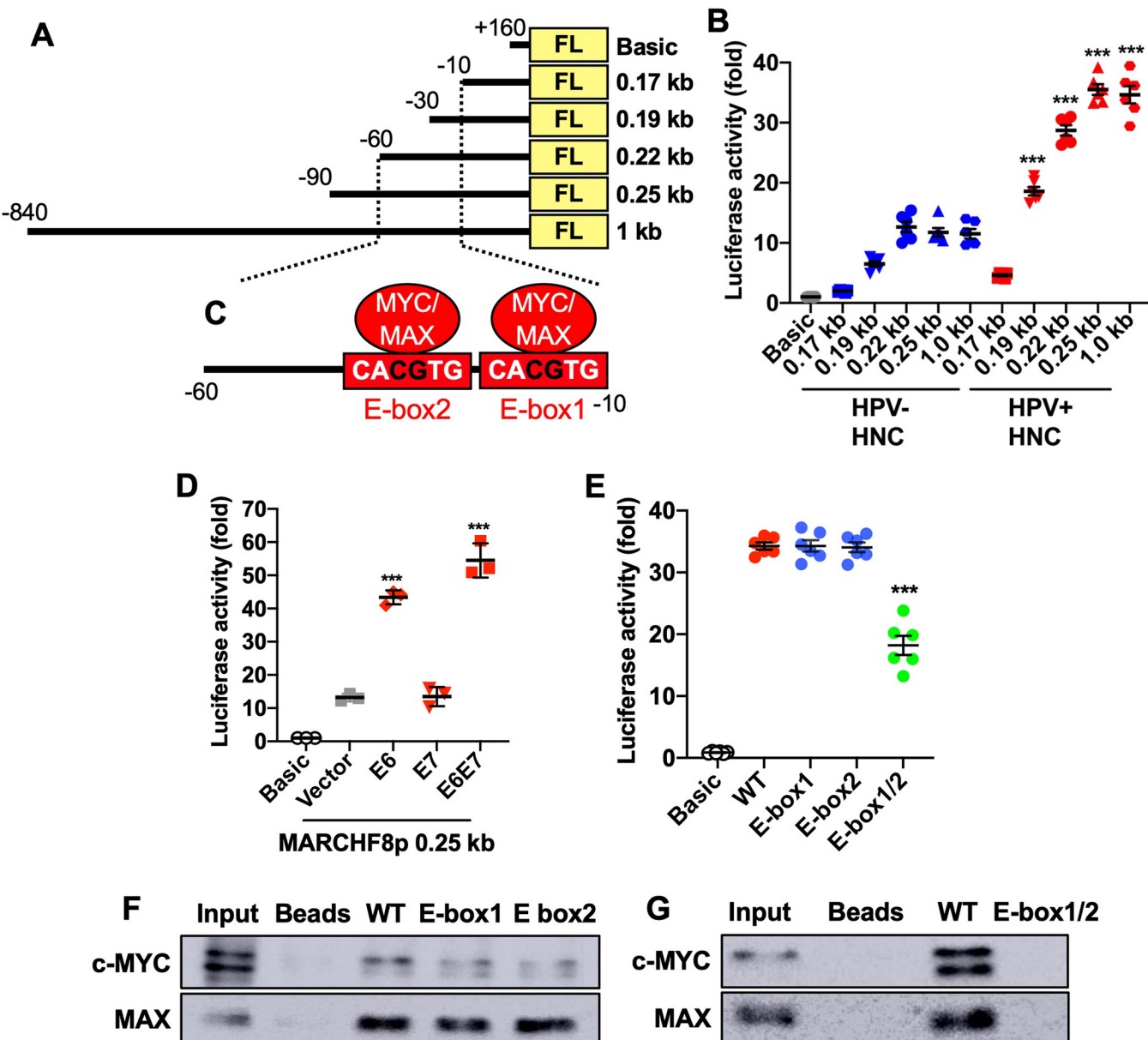

**Fig 2. The HPV oncoprotein E6 induces the *MARCHF8* promoter activity mediated by the MYC/MAX complex.** Schematic representation of the *MARCHF8* promoter regions (-840 to +160) in the firefly luciferase (FL) reporter plasmid pGL4.2 (**A**) and two E-boxes (**C**). The promoter-reporter constructs were transfected into HPV+ (SCC152) and HPV- (SCC1) cells (**B**) and cotransfected into SCC1 cells with HPV16 E6, E7, or E6 and E7 expression plasmids (**D**). SCC152 cells were transfected with the 0.25 kb promoter (-90 to +160) reporter constructs of wildtype (WT) and E-box mutants containing single or double CG deletion in E-box1, E-box-2, and E-box1/2 (**B** and **E**). Luciferase activity was measured 48 h post transfection. Representative data from three independent experiments are shown as a fold change relative to the empty pGL4.2 vector (Basic). Shown are representative data of three repeats. *P* values were determined by Student's *t*-test. $^*p < 0.05$, $^{**}p < 0.01$, $^{***}p < 0.001$. A DNA-protein pulldown assay was performed using biotinylated 90 bp oligonucleotides containing wildtype or E-boxes mutants (-85 to +5) incubated with nuclear extracts from SCC152 cells. The DNA-bound proteins were analyzed using anti-c-MYC and MAX antibodies (**F** and **G**) by western blotting.

To identify the cellular factors that bind to the essential *MARCHF8* promoter regions (-60 to -10) in the context of HPV16 E6, we performed a DNA-protein pulldown assay using biotinylated 90 bp oligonucleotides containing the promoter sequence between -85 and +5. Our liquid chromatography-mass spectrometry identified 83 *MARCHF8* promoter-binding proteins (S2 Table). Our in silico analysis using the Eukaryotic Promoter Database (epd.epfl.ch)

predicted two enhancer boxes (E-box), the known binding sites of the MYC/MAX complex, between -60 and -10 in the *MARCHF8* promoter (**Fig 2C and 2D**). As predicted in our in-silico analysis (**Figs 2C and S3A**), MYC-associated factor X (MAX), a member of the MYC family of transcription factors, was identified as a *MARCHF8* promoter binding protein. The MYC/MAX heterodimer complex binds to the DNA sequence designated E-box (CACGTG). To determine if the E-boxes are necessary for *MARCHF8* promoter activity, we generated a pGL4.2 reporter plasmid with a mutant 0.25 kb promoter region by deleting CG in either or both E-boxes (**Fig 2C**) and determined the promoter activity. While CG deletion in only one of the two E boxes did not change any promoter activity, CG deletions in both E-boxes significantly abrogated the *MARCHF8* promoter (**Fig 2E**). To further validate the binding of MYC and MAX proteins to the E-boxes in the *MARCHF8* promoter, *MARCHF8* promoter binding proteins were pulled down using biotinylated 90 bp oligonucleotides (-85 to +5) containing the wildtype or mutant E-boxes. Then, MYC and MAX proteins were detected by western blotting. Consistent with its promoter activity (**Fig 2E**), while the 90 bp oligonucleotide fragment with CG deletion in only one of the two E-boxes still binds to both MYC and MAX proteins (**Fig 2F**), CG deletion in both E-boxes completely abrogates MYC and MAX binding to the *MARCHF8* promoter (**Fig 2G**). HPV E6 is known to interact with and stabilize the MYC/MAX complex to activate the hTERT promoter [35,36]. Thus, our findings suggest that HPV E6-induced MYC/MAX plays an important role in host transcriptional regulations, including *MARCHF8*.

## Expression of death receptors on HPV+ HNC cells is post-translationally downregulated by HPV oncoproteins

MARCHF8 is known to target several membrane proteins for degradation through ubiquitination. A previous study reported that TRAIL-R1, a TNFRSF death receptor, is ubiquitinated by MARCHF8 [24]. Thus, we first determined total protein levels of the TNFRSF death receptors, FAS, TRAIL-R1, and TRAIL-R2, in HPV+ HNC (SCC2, SCC90, and SCC152) and HPV-HNC (SCC1, SCC9, and SCC19) cells comparing to N/Tert-1 cells by western blotting. The results showed that FAS, TRAIL-R1, and TRAIL-R2 protein levels are significantly lower in all HPV+ HNC cells, except TRAIL-R1 in SCC2, compared to N/Tert-1 cells. In contrast, HPV-HNC cells did not show any significant changes in FAS and TRAIL-R1 except in SCC9 cells, while TRAIL-R2 showed consistent downregulation in both HPV+ and HPV- HNC cells (**Fig 3A and 3B**). Next, cell surface expression of FAS, TRAIL-R1, and TRAIL-R2 on the HPV + HNC and HPV- HNC cells was determined by flow cytometry. Consistent with the total protein levels, FAS, TRAIL-R1, and TRAIL-R2 expression on all HPV+ HNC cells, except TRAIL-R1 on SCC2 cells, is significantly decreased compared to N/Tert-1 cells (**Fig 3C–3H**). In contrast, across the HPV- HNC cells, all three death receptors are expressed at variable levels showing no clear trend of surface expression compared to N/Tert-1 cells (**Fig 3C–3H**). Next, we measured mRNA levels of FAS, TRAIL-R1, and TRAIL-R2 in HPV+ and HPV-HNC cells along with N/Tert-1 cells by RT-qPCR. The results showed that FAS mRNA levels are upregulated in HPV+ HNC cells, while FAS mRNA levels are not changed or slightly decreased in HPV- HNC cells compared to N/Tert-1 cells (**S4A Fig**). In addition, mRNA expression of TRAIL-R1 (**S4B Fig**) and TRAIL-R2 (**S4C Fig**) is decreased in HPV- HNC cells and variable in HPV+ HNC cells. These results suggest that the downregulation of FAS, TRAIL-R1, and TRAIL-R2 expression on HPV+ HNC cells is likely caused by post-translational regulation.

Next, we determined whether the HPV16 oncoproteins E6 and E7 contribute to the downregulation of FAS, TRAIL-R1, and TRAIL-R2, using N/Tert-1 cells expressing E6 (N/Tert-1

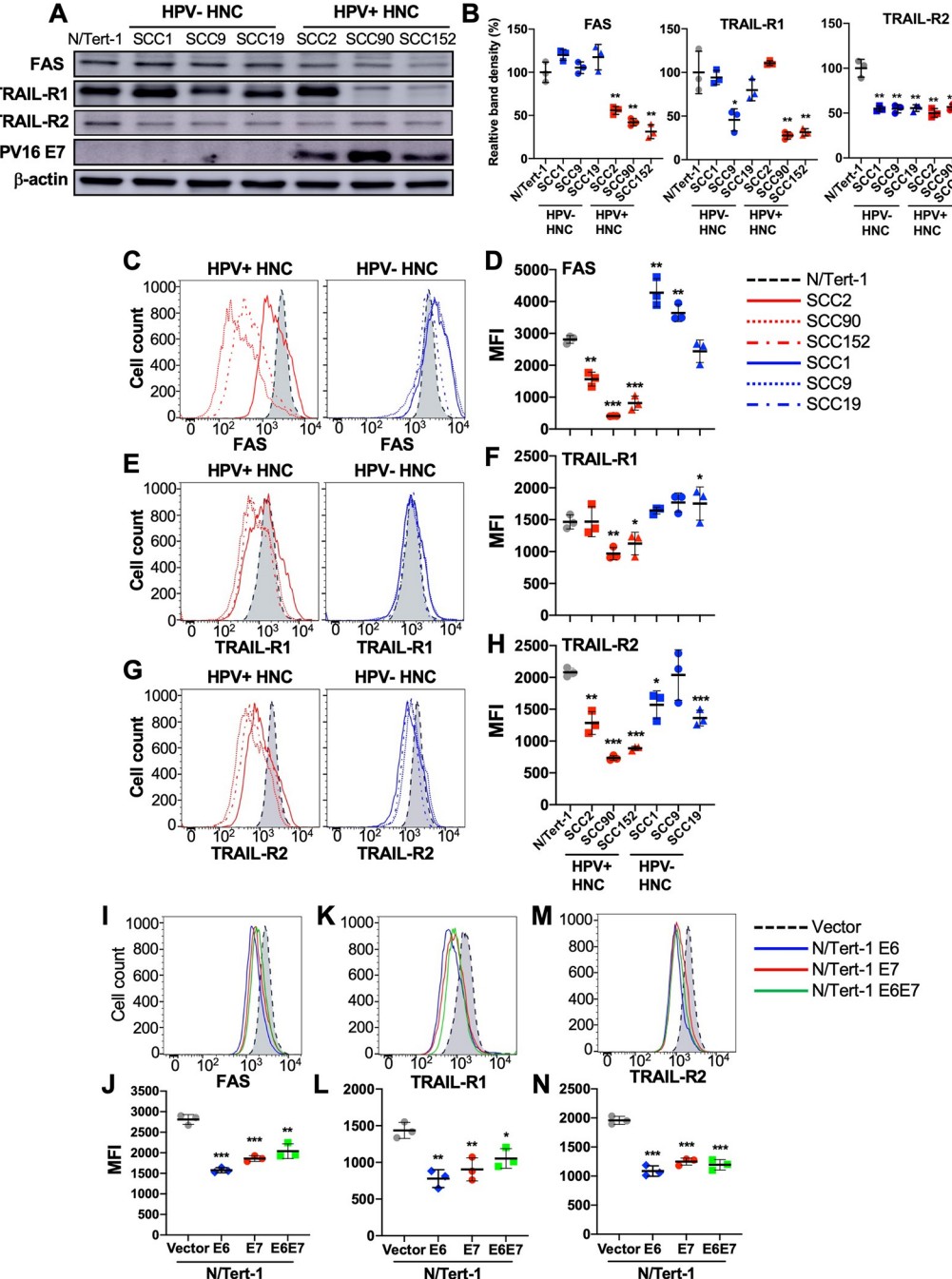

**Fig 3. Expression of FAS, TRAIL-R1, and TRAIL-R2 is downregulated in HPV+ HNC cells.** Total protein expression of FAS, TRAIL-R1, and TRAIL-R2 in HPV- (SCC1, SCC9, and SCC19) and HPV+ (SCC2, SCC90, and SCC152) HNC cells were determined by western blotting (**A**). Relative band density was quantified using NIH ImageJ (**B**). HPV16 E7 and β-actin were used as viral and internal controls, respectively. Cell surface expression of FAS (**C** and **D**), TRAIL-R1 (**E** and **F**), and TRAIL-R2 (**G** and **H**) proteins on HPV+ (SCC2, SCC90, and SCC152) and HPV- (SCC1, SCC9, and SCC19) HNC cells was analyzed by flow cytometry. Cell surface expression of FAS (**I** and **J**), TRAIL-R1 (**K** and **L**), and TRAIL-R2 (**M** and **N**) proteins on N/Tert-1 cells expressing HPV16 E6, E7, and E6E7 was analyzed by flow cytometry. Mean fluorescence intensities (MFI) of three independent experiments are shown (**D**, **F**, **H**, **J**, **L**, and **N**). All experiments were repeated at least three times, and the data shown are means ± SD. $P$ values were determined by Student's $t$-test. $^*p < 0.05$, $^{**}p < 0.01$, $^{***}p < 0.001$.

E6), E7 (N/Tert-1 E7), or both E6 and E7 (N/Tert-1 E6E7). Interestingly, our data showed that expression of either or both HPV16 E6 and E7 is sufficient for a significant decrease in FAS, TRAIL-R1, and TRAIL-R2 expression on N/Tert-1 cells (**Fig 3I–3N**). Additionally, we determined mRNA levels of FAS, TRAIL-R1, and TRAIL-R2 in N/Tert-1 cells expressing E6 (N/Tert-1 E6), E7 (N/Tert-1 E7), or both E6 and E7 (N/Tert-1 E6E7) by RT-qPCR. The results showed that FAS mRNA levels are upregulated in N/Tert-1 E6 and N/Tert-1 E6E7 cells, while is slightly decreased significantly in N/Tert-1 E7 cells compared to N/Tert-1 cells (**S4D Fig**). On the other hand, mRNA levels of TRAIL-R1 decreased significantly in all three N/Tert-1 cells expressing HPV16 E6 and/or E7 (**S4E Fig**), while TRAIL-R2 mRNA expression is upregulated in N/Tert-1 E6 and downregulated in N/Tert-1 E7 cells (**S4F Fig**). Our results suggest that surface expression of the TNFRSF death receptors on HPV+ HNC cells is post-translationally downregulated by the HPV oncoproteins E6 and E7.

## Knockdown of *MARCHF8* expression increases FAS, TRAIL-R1, and TRAIL-R2 expression in HPV+ HNC cells

To determine if the *MARCHF8* upregulation by E6 and E7 is responsible for the downregulation of FAS, TRAIL-R1, and TRAIL-R2 in HPV+ HNC cells, we knocked down *MARCHF8* expression in SCC152 cells using five unique shRNAs against *MARCHF8* (shR-MARCHF8, clones 1–5) delivered by lentiviruses and selected by puromycin treatment. All five shR-MARCHF8s in SCC152 cells showed a ~50% decrease in total MARCHF8 protein levels compared to the control SCC152 cells with nonspecific scrambled shRNA (shR-scr) (**Fig 4A and 4B**). Using western blotting and flow cytometry, we analyzed the protein expression of FAS, TRAIL-R1, and TRAIL-R2. We found that both total (**Fig 4A and 4B**) and cell surface (**Fig 4C–4H**) protein levels of FAS, TRAIL-R1, and TRAIL-R2 are significantly increased by all five cell lines with *MARCHF8* knockdown. We also knocked down *MARCHF8* expression in another HPV+ HNC cell line, SCC2, using three shRNAs and confirmed the increase of total (**S5A and S5B Fig**) and surface (**S5C–S5H Fig**) FAS, TRAIL-R1, and TRAIL-R2 expression by *MARCHF8* knockdown. To determine whether knockdown of *MARCHF8* expression affects mRNA expression of FAS, TRAIL-R1, and TRAIL-R2, we measured mRNA levels of FAS, TRAIL-R1, and TRAIL-R2 in the SCC152 and SCC2 cells with *MARCHF8* knockdown by RT-qPCR. The results showed no significant changes in FAS mRNA expression by *MARCHF8* knockdown in both SCC152 (**S6A Fig**) and SCC2 cells (**S6D Fig**). Additionally, mRNA expression of TRAIL-R1 and TRAIL-R2 is decreased in SCC152 cells (**S6B and S6C Fig**) but not in SCC2 cells (**S6E and S6F Fig**) with *MARCHF8* knockdown, indicating that the decrease of FAS, TRAIL-R1, and TRAIL-R2 protein levels in HPV+ HNC cells are not caused by a reduction in their mRNA levels. These results suggest that HPV oncoprotein-induced MARCHF8 post-translationally downregulates the TNFRSF death receptors.

## MARCHF8 protein interacts with and ubiquitinates FAS, TRAIL-R1, and TRAIL-R2 proteins in HPV+ HNC cells

Previous studies have shown that MARCHF8 binds to and ubiquitinates several membrane receptor proteins for degradation [24,37,38]. Thus, we hypothesized that MARCHF8 upregulated by the HPV oncoproteins targets FAS, TRAIL-R1, and TRAIL-R2 proteins for ubiquitination and degradation. First, to determine if MARCHF8 protein binds to FAS, TRAIL-R1, and TRAIL-R2, we pulled down MARCHF8 protein in whole cell lysates from SCC152 cells treated with the proteasome inhibitor MG132 using magnetic beads conjugated with an anti-MARCHF8 antibody. The western blot analyses showed that FAS, TRAIL-R1, and TRAIL-R2 proteins were detected in the MARCHF8 protein complex pulled down with an anti-

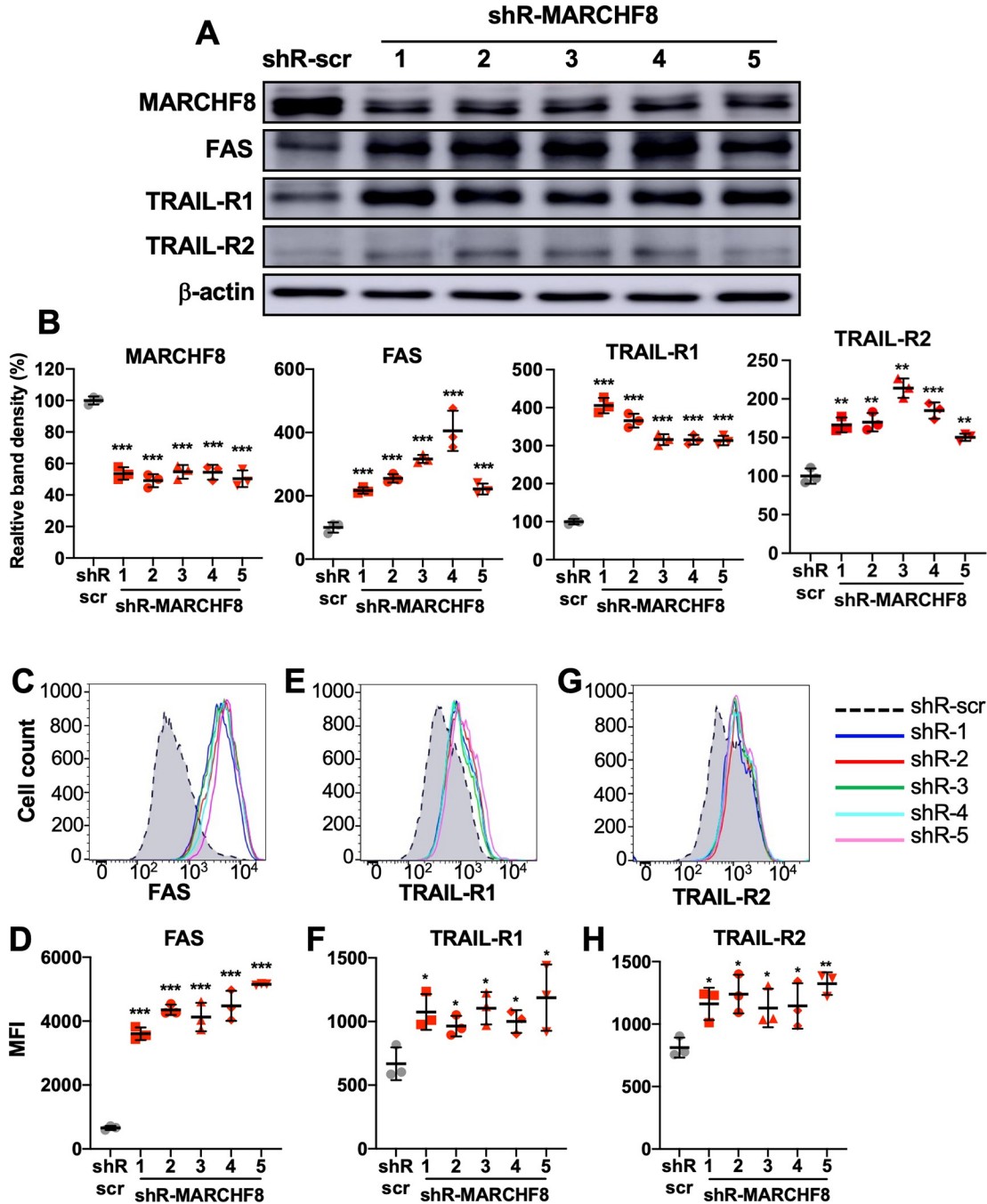

**Fig 4. Knockdown of *MARCHF8* expression increases FAS, TRAIL-R1, and TRAIL-R2 protein expression in HPV+ HNC cells.** HPV+ HNC (SCC152) cells were transduced with one of five lentiviral shRNAs against *MARCHF8* (shR-MARCHF8 clones 1–5) or scrambled shRNA (shR-scr) as a control. Protein expression of MARCHF8, FAS, TRAIL-R1, and TRAIL-R2 was determined by western blotting (**A**). Relative band density was quantified using NIH ImageJ (**B**). β-actin was used as an internal control. The data shown are means ± SD of three independent experiments. Cell surface expression of FAS (**C** and **D**), TRAIL-R1 (**E** and **F**), and TRAIL-R2 (**G** and **H**) proteins were analyzed by flow cytometry. Mean fluorescence intensities (MFI) of three independent experiments are shown (**D**, **F**, and **H**). *P* values were determined by Student's *t*-test. $^{*}p < 0.05$, $^{**}p < 0.01$, $^{***}p < 0.001$.

MARCHF8 antibody (**Fig 5A**). Reciprocally, the co-immunoprecipitation of FAS protein in the same whole cell lysates using an anti-FAS antibody showed MARCHF8 protein (**Fig 5B**). Next, to determine if MARCHF8 induces ubiquitination of FAS, TRAIL-R1, and TRAIL-R2 proteins, we pulled down ubiquitinated proteins in whole-cell lysates from SCC152 cells treated with MG132 using magnetic beads conjugated with an anti-ubiquitin antibody. The results showed that the levels of ubiquitinated FAS (**Figs 5C, 5F, S7A and S7B**), TRAIL-R1 (**Figs 5D, 5G, S7C and S7D**), and TRAIL-R2 (**Figs 5E, 5H, S7E and S7F**) proteins were decreased in SCC152 cells by *MARCHF8* knockdown, despite the significantly higher levels of total input proteins of FAS, TRAIL-R1 and TRAIL-R2, compared to SCC152 cells with shR-scr. These results suggest that HPV-induced MARCHF8 binds to and ubiquitinates the TNFRSF death receptors in HPV+ HNC cells.

## Knockdown of *MARCHF8* expression increases apoptosis of HPV+ HNC cells

Since expression of the death receptors is significantly upregulated by *MARCHF8* knockdown (**Figs 4 and S5**), we hypothesized that the HPV oncoproteins inhibit host cell apoptosis by inducing MARCHF8-mediated degradation of FAS, TRAIL-R1, and TRAIL-R2 proteins. To test this hypothesis, we determined whether the knockdown of *MARCHF8* expression in HPV + HNC cells enhances FAS-mediated apoptosis. SCC152 and SCC2 cells with two shR-MARCHF8 or shR-scr were first sensitized for apoptosis by treating with an anti-human FAS antibody or the soluble recombinant FAS ligand (rFAS-L), and apoptotic cells were quantified by detecting annexin V- and 7-aminoactinomycin D (7-AAD)-positive cells. The results showed that both SCC152 (**Fig 6A and 6B**) and SCC2 (**Fig 6C and 6D**) cells with *MARCHF8* knockdown displayed significantly increased percentages of apoptotic cells compared to the corresponding cells with shR-scr under the sensitization with anti-FAS antibody or rFAS-L. These results suggest that by ubiquitinating and degrading FAS proteins, MARCHF8 is transcriptionally upregulated by HPV oncoproteins and inhibits host cell apoptosis.

## *Marchf8* expression is upregulated, and death receptor expression is downregulated in HPV+ mouse oral cancer cells

To further investigate the inhibition of apoptosis by HPV-induced MARCHF8 degradation of the TNFRSF death receptors, we adopted a mouse model of HPV+ HNC with *Hras*-transformed mouse oral epithelial cells expressing HPV16 E6 and E7 (MOE/E6E7 or mEERL) that form tumors in immunocompetent syngeneic C57BL/6 mice [39]. First, we determined total protein levels of MARCHF8, FAS, TRAIL-R1, and TRAIL-R2 in mEERL cells compared to normal immortalized mouse oral epithelial (NiMOE) cells and mouse MOE cells transformed with *Hras* and shR-Ptpn14 (HPV- MOE). The results from the mouse oral cancer cells were consistent with those from human HNC cells showing a significant increase in MARCHF8 protein levels and a decrease in FAS, TRAIL-R1, and TRAIL-R2 protein levels in mEERL cells compared to NiMOE cells (**Fig 7A and 7B**). Similarly, surface expression of FAS and TRAIL-R2 proteins was significantly downregulated in mEERL cells compared to NiMOE cells (**Fig 7C–7F**). While there were no significant changes in TRAIL-R2 protein levels in HPV- MOE cells compared to NiMOE cells (**Fig 7A, 7B, 7E and 7F**), total and cell surface expression of FAS and TRAIL-R1 proteins were decreased in HPV- MOE cells compared to NiMOE cells (**Fig 7B–7D**). These results suggest that mEERL cells recapitulate our findings from human HPV+ HNC cells that HPV-induced MARCHF8 degrades the TNFRSF death receptors.

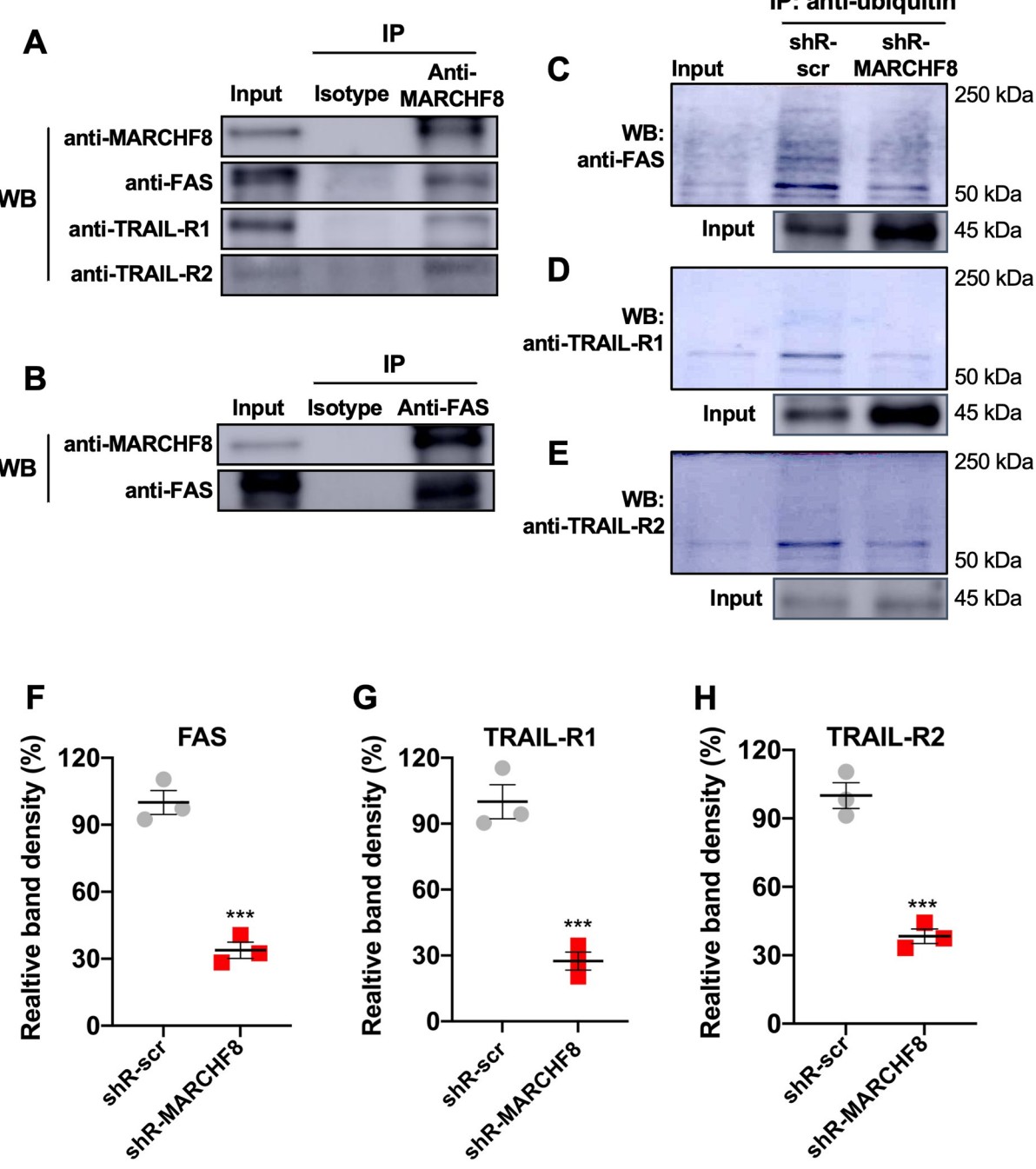

**Fig 5. MARCHF8 protein interacts with and ubiquitinates FAS, TRAIL-R1, and TRAIL-R2 proteins.** MARCHF8 (**A**) and FAS (**B**) were pulled down from the cell lysate of HPV+ HNC (SCC152) cells treated with a proteasome inhibitor MG132 using anti-MARCHF8 (**A**) and anti-FAS (**B**) antibodies, respectively. Western blotting detected FAS, TRAIL-R1, TRAIL-R2, and MARCHF8 proteins in the immunoprecipitated proteins. Ubiquitinated proteins were pulled down from the cell lysate of HPV+ HNC (SCC152) cells with scrambled shRNA (shR-scr) or shRNA against MARCHF8 (shR-MARCHF8 clone 3) treated with a proteasome inhibitor MG132 using an anti-ubiquitin antibody (**C—H**). FAS (**C and F**), TRAIL-R1 (**D and G**), and TRAIL-R2 (**E and H**) proteins were detected in the immunoprecipitated proteins by western blotting. Relative band density was quantified using ImageJ. The band densities of the shR-MARCHF8 are normalized to the shR-scr. The data shown are means ± SD from three repeats (**Figs 5C–5E and S7A–S7F**). $P$ values were determined by Student's $t$-test. $^{*}p < 0.05$, $^{**}p < 0.01$, $^{***}p < 0.001$.

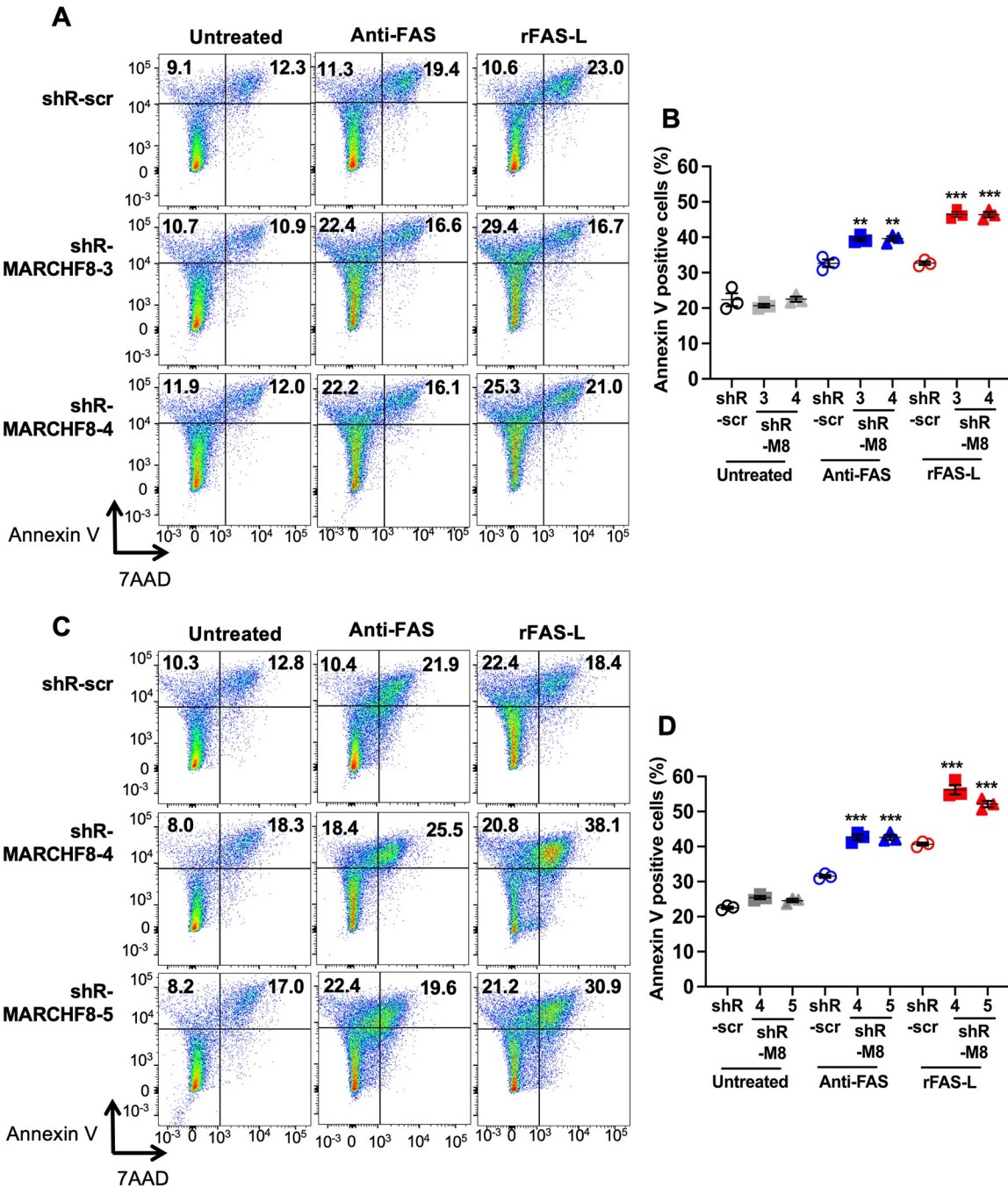

**Fig 6. Knockdown of *MARCHF8* enhances apoptosis of HPV+ HNC cells.** Two HPV+ HNC cells, SCC152 (**A** and **B**) and SCC2 (**C** and **D**), with scrambled shRNA (shR-scr) or two shRNAs against MARCHF8 (shR-MARCHF8), were treated with an anti-human FAS antibody (Anti-FAS, clone EOS9.1, eBioscience) or the recombinant FAS ligand (rFAS-L, BioLegend #585404). The cells were stained with an anti-annexin V antibody and 7-AAD and analyzed by flow cytometry. The percentage of cells with positive staining is indicated (**A** and **C**). The data shown are means ± SD of three independent experiments (**B** and **D**). *P* values were determined by Student's *t*-test. $^{**}p < 0.01$, $^{***}p < 0.001$.

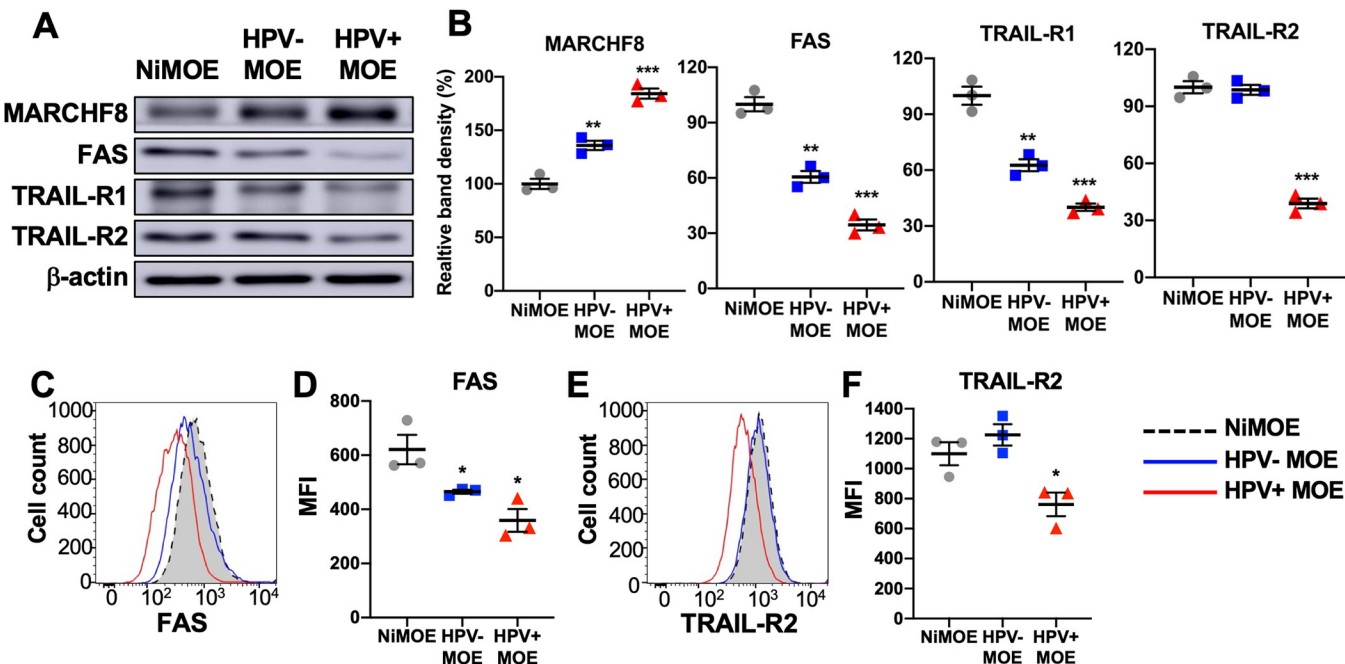

**Fig 7. *MARCHF8* is upregulated, and death receptor expression is downregulated in HPV+ mouse oral cancer cells.** Mouse MARCHF8, FAS, TRAIL-R1, and TRAIL-R2 protein levels in mouse normal immortalized (NiMOE), HPV- transformed (HPV- MOE), and HPV+ transformed (HPV+ MOE) oral epithelial cells were determined by western blotting (**A**). β-actin was used as a loading control. The relative band density was quantified using NIH ImageJ (**B**). Cell surface expression of FAS (**C and D**) and TRAIL-R2 (**E and F**) proteins on NiMOE (dotted black line), HPV- MOE (blue line), and HPV+ MOE (red line) cells were analyzed by flow cytometry. Mean fluorescence intensities (MFI) of three independent experiments are shown (**D and F**). All experiments were repeated at least three times, and the data shown are means ± SD. *P* values were determined by Student's *t*-test. $^*p < 0.05$, $^{***}p < 0.001$.

### *Marchf8* knockout in HPV+ mouse oral cancer cells restore FAS, TRAIL-R1, and TRAIL-R2 expression and enhances apoptosis

To determine if the high levels of *Marchf8* expression in HPV+ MOE cells are responsible for the downregulation of FAS, TRAIL-R1, and TRAIL-R2, we established *Marchf8* knockout mEERL (mEERL/*Marchf8*$^{-/-}$) cell lines using lentiviral transduction of Cas9 and three small guide RNAs (sgRNAs) against *Marchf8* (sgR-*Marchf8*, clones 1–3). mEERL cells transduced with Cas9 and scrambled sgRNA (mEERL/scr) were used as a control. mEERL cell lines transduced with two (clones 2 and 3) of the three sgR-*Marchf8*s showed a ~75% decrease in *Marchf8* expression compared to mEERL/scr cells (**Fig 8A and 8B**). Consistent with the data from human HPV+ HNC cells presented in **Fig 4**, mEERL/*Marchf8*$^{-/-}$ cells showed significantly upregulated total FAS, TRAIL-R1, and TRAIL-R2 protein levels compared to mEERL/scr cells (**Fig 8A and 8B**). We also detected upregulated surface expression of FAS, TRAIL-R1, and TRAIL-R2 on mEERL/*Marchf8*$^{-/-}$ cells compared to mEERL/scr cells (**Fig 8C–8F**). We could not examine surface expression of TRAIL-R1, as a fluorophore-conjugated mouse TRAIL-R1 antibody for flow cytometry was unavailable. Next, we determined whether *Marchf8* knockout enhances apoptosis of mEERL/*Marchf8*$^{-/-}$ cells by sensitizing with mouse rFAS-L. The results showed that mEERL/*Marchf8*$^{-/-}$ cells showed significantly increased annexin V- and 7-AAD-positive cells compared to mEERL/scr cells (**Fig 8G and 8H**). These results are consistent with our findings in human HPV+ HNC that HPV-induced *MARCHF8* expression inhibits host cell apoptosis by degrading the TNFRSF death receptors.

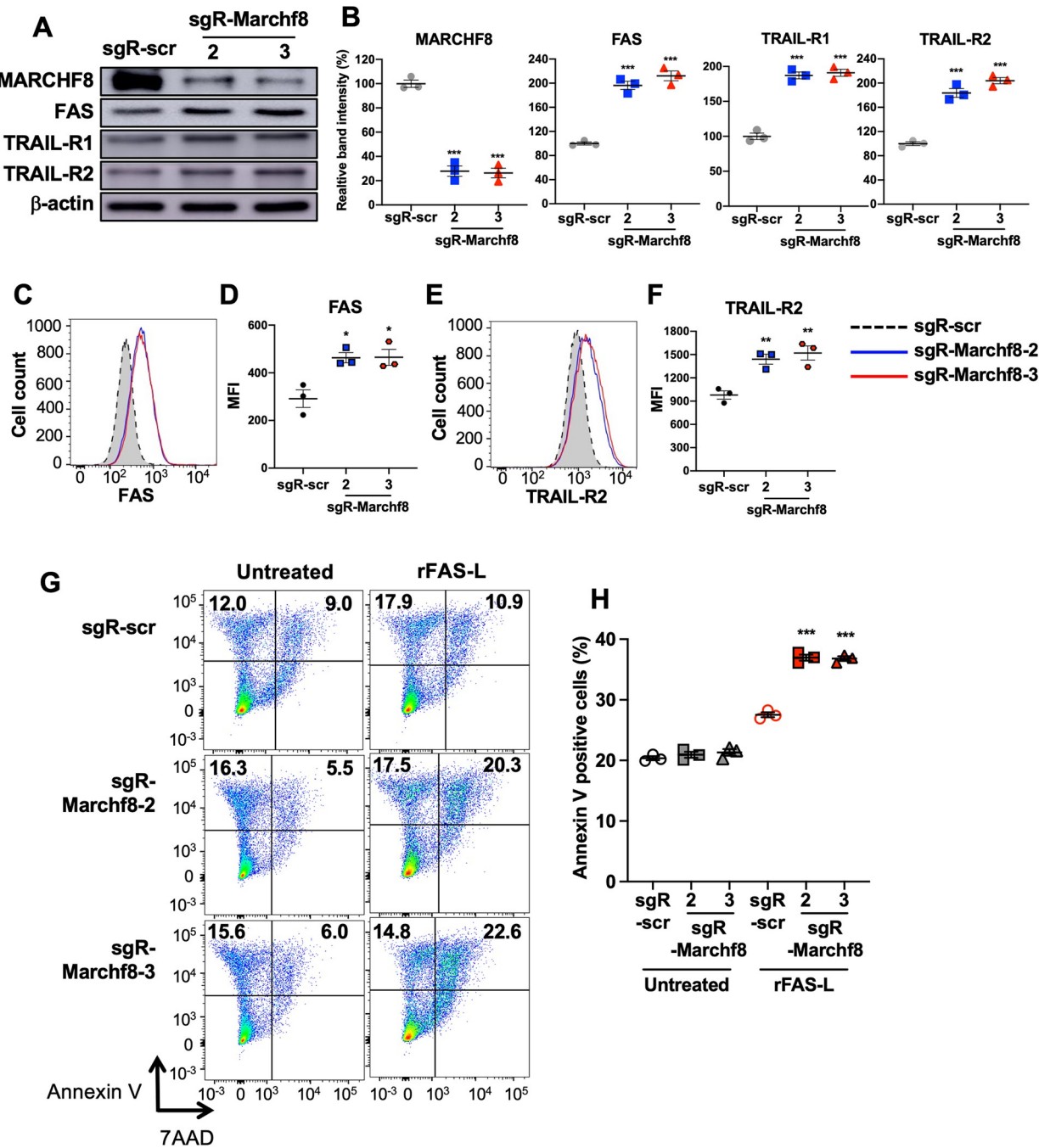

**Fig 8. Knockout of *Marchf8* expression increases FAS, TRAIL-R1, and TRAIL-R2 protein levels and enhances apoptosis of HPV+ mouse oral cancer cells.** mEERL cells were transduced with lentiviral Cas9 and one of two sgRNAs against Marchf8 (sgR-Marchf8-2 and sgR-Marchf8-3) or scrambled sgRNA (sgR-scr). Protein levels of MARCHF8, FAS, TRAIL-R1, and TRAIL-R2 were determined by western blotting (**A**). Relative band density was quantified using NIH ImageJ (**B**). β-actin was used as a loading control. The data shown are means ± SD of three independent experiments. Cell surface expression of FAS (**C** and **D**) and TRAIL-R2 (**E** and **F**) proteins were analyzed by flow cytometry. Mean fluorescence intensities (MFI) of three independent experiments are shown (**D** and **F**). *P* values were determined by Student's *t*-test. $^{*}p < 0.05$, $^{**}p < 0.01$, $^{***}p < 0.001$. Untreated and rFAS-L-treated mEERL cells (**G**) with sgR-Marchf8-2, sgR-Marchf8-3, or sgR-scr were stained with an anti-annexin V antibody and 7-AAD and analyzed by flow cytometry. The percentage of cells with positive staining is indicated (**G**). The data shown are means ± SD of three independent experiments (**H**). *P* values were determined by Student's *t*-test. $^{***}p < 0.001$.

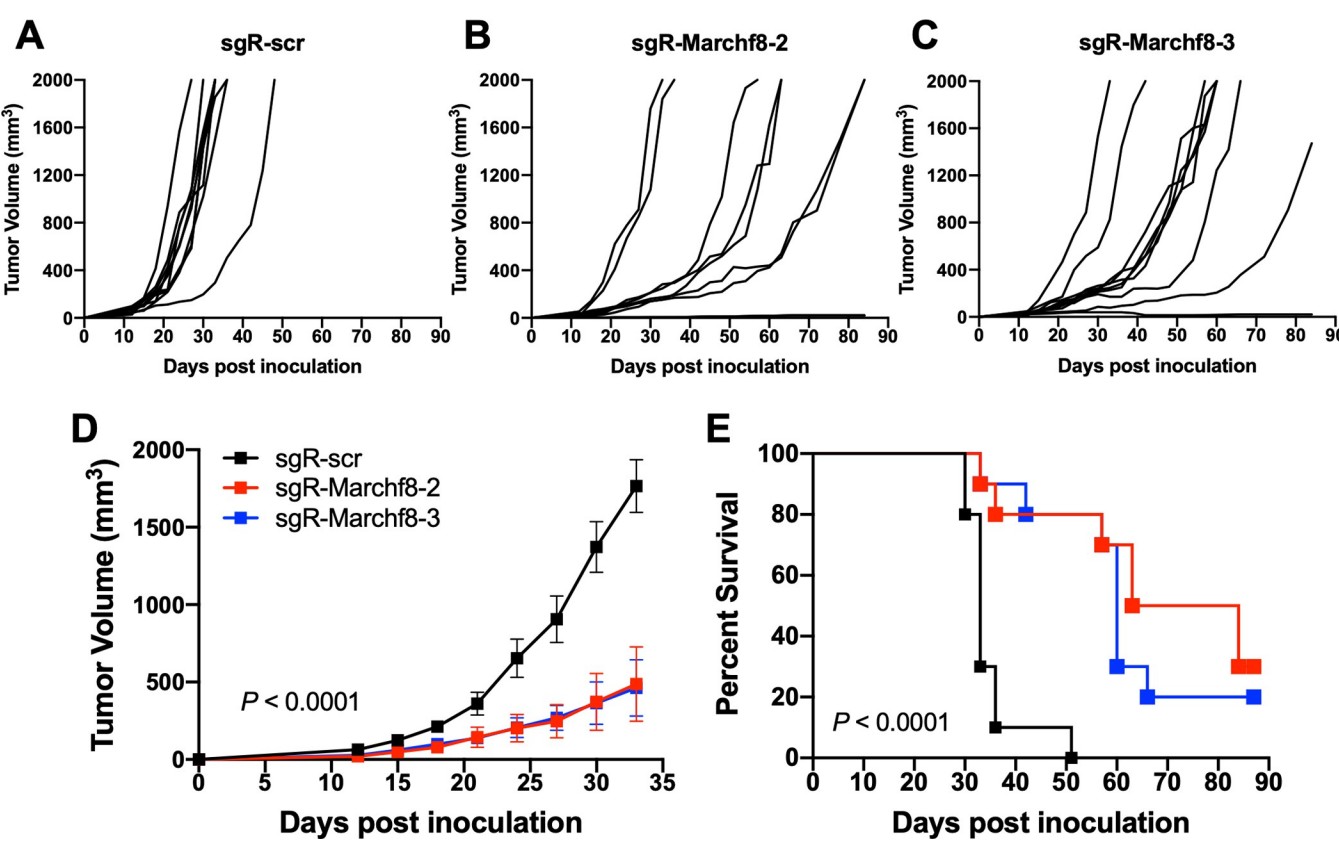

**Fig 9. Knockout of *Marchf8* expression suppresses HPV+ HNC tumor growth in vivo.** *Marchf8*-knockout mEERL cells (mEERL/*Marchf8*^-/-) were generated by lentiviral Cas9, and two sgRNAs targeting *Marchf8* (sgR-Marchf8-2 and sgR-Marchf8-3) or scrambled sgRNA (sgR-scr). mEERL/scr (**A**) or mEERL/*Marchf8*^-/- (**B** and **C**) cells were injected into the rear right flank of C57BL/6J mice (*n* = 10 per group). Tumor volume was measured twice a week (**A-D**). Survival rates of mice were analyzed using a Kaplan-Meier estimator (**E**). The time to event was determined for each group, with the event defined as a tumor size larger than 2000 mm³. The data shown are means ± SD. *P* values of mice injected with mEERL/*Marchf8*^-/- cells compared with mice injected with mEERL/scr cells were determined for tumor growth (**D**) and survival (**E**) by two-way ANOVA analysis. Shown are representative of two independent experiments.

## *Marchf8* knockout in HPV+ HNC cells suppresses tumor growth in vivo

To determine whether *Marchf8* knockout in HPV+ HNC suppresses tumor growth in vivo, we injected syngeneic C57BL/6J mice subcutaneously with either of two 5 X 10⁵ of mEERL/*Marchf8*^-/- cell lines or mEERL/scr cells into the flank. Tumor growth was monitored twice a week for 12 weeks. All ten mice injected with mEERL/scr cells showed vigorous tumor growth (**Fig 9A and 9D**) and succumbed to tumor burden ~7 weeks post injection (**Fig 9E**). In contrast, only two out of ten mice, each injected with either of two mEERL/*Marchf8*^-/- cell lines, showed robust tumor growth (**Fig 9B, 9C, and 9E**) and died 8 weeks post injection (**Fig 9E**). Further, no tumor formation was observed in three and one mice injected with mEERL/*Marchf8*^-/- cell lines, sgR-Marchf8-2 and sgR-Marchf8-3, respectively, over 12 weeks post injection (**Fig 9D and 9E**). Our results suggest that MARCHF8 is a potent tumor promoter that plays an important role in cancer progression by inducing the degradation of the TNFRSF death receptors and blocking cell apoptosis.

## Discussion

As cancer cells must prevent apoptosis for survival, the TNFRSF death receptors are frequently dysregulated in cancer cells [40]. To inhibit death receptor-mediated apoptosis, cancer cells

often repress *FAS* expression by histone modification [41], polymorphism [42], and hyper-methylation of the promoter region [43,44]. Cancer cells also abrogate the function of the death receptors by generating loss-of-function mutations [45,46] and inducing the expression of inhibitory molecules such as FLICE-like inhibitory protein [47–49], decoy receptors and ligands [50,51], and microRNA miR-196b [52,53]. Additionally, cancer cells dysregulate the trafficking of death receptors to interfere with their cell surface expression [54,55]. Interestingly, MARCHF8 ubiquitinates TRAIL-R1 and diminishes its cell surface expression on breast cancer cells [24]. In addition, a previous study has shown that proteasome inhibitor treatment upregulates TRAIL-R2 protein and induces apoptosis in prostate cancer cells [56]. These findings suggest that inhibiting TNFRSF death receptors-mediated apoptosis through ubiquitination plays an important role in cancer cell survival. Nevertheless, the detailed mechanisms by which HPV+ HNC cells inhibit TNFRSF death receptors were mostly unknown.

This study reports that expression of the E3 ubiquitin ligase MARCHF8 is upregulated by HPV oncoproteins, downregulating surface expression of FAS, TRAIL-R1, and TRAIL-R2 on HPV+ HNC cells (**Fig 10**). The TNFSFR death receptors, characterized by a cytoplasmic tail termed the death domain, play a crucial role in apoptosis upon their interactions with specific extracellular ligands such as FAS-L and TRAIL (a.k.a. TNFSF10) [57–59]. FAS, a member of the TNFRSF expressed in various tissues, interacts with FAS-L (a.k.a. CD178) to trigger apoptosis [60,61]. TRAIL-R1 and TRAIL-R2 interact with TRAIL (a.k.a. TNFSF10) [62,63]. In contrast, the other TRAIL-Rs, TRAIL-R3 and TRAIL-R4, do not have any death domain and are not considered death receptors.

MARCHF8, originally named cellular MIR (c-MIR), was first discovered as a human homolog of two KSHV proteins, a modulator of immune recognition 1 and 2 (MIR1 and MIR2, a.k.a. K3 and K5, respectively) [17]. Like KSHV MIR1 and MIR2, MARCHF8 downregulates the surface expression of various immunoreceptors, including MHC-II [64], CD44 [23,65], CD81 [65], and CD86 [66]. MARCHF8 also decreases the surface expression of TRAIL-R1 (21, 58) and inhibits apoptosis. In addition to KSHV, many other viruses employ various strategies to restrain host cell apoptosis, which is considered an antiviral host innate response [67]. For example, DNA viruses such as adenovirus encode viral proteins that downregulate death receptors and inhibit caspase activation [68]. It has been suggested that some of these anti-apoptotic functions of the viruses contribute to the oncogenic process during virus-driven cancer progression[68–71].

High-risk HPVs have been shown to inhibit host cell apoptosis by targeting several different apoptotic mechanisms, especially the TNFRSF death receptors and their signaling [72]. While HPV E5 is involved in impeding TNFRSF-mediated apoptosis [73], E6 plays a crucial role in the inhibition of host cell apoptosis by interacting with TNFRSF1A and FAS-associated death domain (FADD) and facilitating their degradation [15,74]. Furthermore, the knockdown of E6 expression or treatment with a proteasome inhibitor significantly enhances TNFRSF death receptor-mediated apoptosis of cervical cancer cells [75,76]. These results indicate that E6-mediated inhibition of apoptosis through TNFRSF death receptors is critical for cancer cell survival. However, the mechanism of E6-mediated degradation of the TNFRSF death receptors was mostly elusive. Our study has revealed that HPV16 E6 induces the degradation of the TNFRSF death receptors by activating the *MARCHF8* promoter through the interaction of the MYC/MAX complex (**Fig 2**), a well-known oncogenic transcription factor complex that also activates hTERT transcription [36,77]. Our results show that E6 mutants deficient in p53 and E6AP binding are still capable of inducing the *MARCHF8* promoter and upregulating *MARCHF8* expression (**S2** and **S3B** Figs). Consistently, Liu et al. previously showed that MYC overexpression could replace HPV E6 to immortalize keratinocytes in the presence of HPV E7 despite the lack of p53 degradation [78]. In addition, E6 stabilizes MYC by enhancing its O-

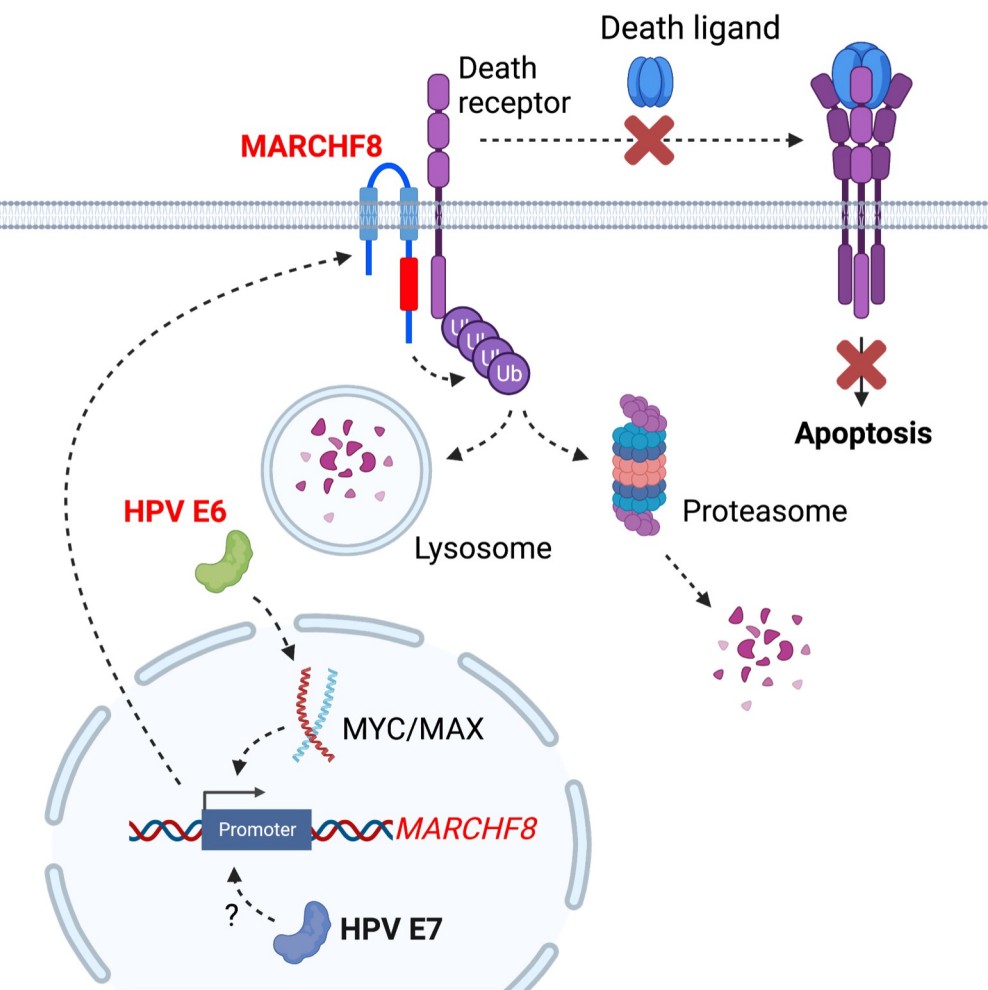

**Fig 10. The schematic diagram summarizes that HPV E6-induced *MARCHF8* expression inhibits host cell apoptosis by degrading the TNFRSF death receptors.** The HPV oncoprotein E6 activates the *MARCHF8* promoter activity through the MYC/MAX transcription factor complex (**A**) and upregulates cell surface expression of the MARCHF8 protein (**B**). MARCHF8 protein binds to and ubiquitinates the TNFRSF death receptors FAS, TRAIL-R1, and TRAIL-R2 (**C**). Ubiquitinated TNFRSF death receptors may be degraded by lysosomes (**D**) or proteasomes in the cytoplasm (**E**). Created with BioRender.com.

linked GlcNAcylation [35], suggesting that E6-induced MYC plays an important role in E6-mediated oncogenesis. While hTERT induction by MYC is crucial, our findings indicate that *MARCHF8* activation by MYC may also play an essential role in cell immortalization by inhibiting apoptosis. However, as MYC is also involved in p53-induced apoptosis [79], the E6 function in p53 degradation may still be necessary for cancer progression. In addition, our results show that keratinocytes expressing HPV16 E7 alone also have increased levels of MARCHF8 mRNA and protein (**Fig 1C**). However, E7 expression alone does not induce the immediate upstream promoter (**Figs 2D and S3B**), suggesting that E7 may upregulate *MARCHF8* expression through an alternative mechanism such as distal enhancer activation and/or mRNA stabilization. Notably, the difference in expression of MARCHF8 mRNA between HPV+ and HPV- HNCs is relatively small, and MARCHF8 mRNA levels in HPV-HNC patients are highly variable compared to MARCHF8 mRNA levels in HPV+ HNC patients (**Fig 1A**). This may imply that *MARCHF8* expression in HPV- HNC might be regulated by various factors, while MARCHF8 expression is upregulated mainly by HPV,

suggesting that different mechanisms may mediate the upregulation of MARCHF8 in HPV + and HPV- HNCs.

It has been discovered that *MARCHF8* expression is upregulated in gastric and esophageal cancer [25], and that its expression is associated with poor prognosis [80,81]. In addition, MARCHF8 ubiquitinates TRAIL-R1 and decreases apoptosis in gastric and breast cancer cells [24,80], and silencing of MARCHF8 induces apoptosis and suppresses cell proliferation, invasion, and migration of cancer cells [25,27]. As described above, MARCHF8 plays a vital role in immune suppression by degrading MHC-II and CD86 [21,22]. Our study shows that *MARCHF8* knockout dramatically suppresses tumor growth in vivo (**Fig 9**). Together, these results suggest that MARCHF8, as a tumor promoter, could be a potential target for cancer therapy to induce cancer cell apoptosis and antitumor immune responses.

Despite these protumor functions in several cancers, including HPV+ HNC, some studies have shown the potential antitumor activity of MARCHF8. Overexpression of *MARCHF8* inhibits NSCLC cell proliferation and metastasis via the PI3K and mTOR signaling pathways [81]. In addition, *MARCHF8* overexpression also promotes apoptosis and hinders tumorigenesis and metastasis of breast cancer cells by downregulating CD44 and STAT3 [82]. These results imply that MARCHF8 may differentially contribute to cancer development and that other MARCHF family members, such as MARCHF1, MARCHF4, and MARCHF9, may have similar functions in the place of MARCHF8. Our study clearly shows the function of MARCHF8 as an oncoprotein in HPV+ HNC. Further investigation is required, as MARCHF8 and other MARCHF family members exhibit extensive roles in the regulation of various membrane proteins involved in cellular homeostasis, apoptosis, and immune responses.

## Materials and methods

### Cell lines

HPV+ (SCC2, SCC90, and SCC152) and HPV- (SCC1, SCC9, and SCC19) HNC cells and 293FT cells were purchased from the American Type Culture Collection and Thermo Fisher, respectively. These cells were cultured and maintained as described [83–86]. The N/Tert-1 cells [87] and their derivatives expressing HPV16 E6 (N/Tert-1-E6), E7 (N/Tert-1-E7), E6 and E7 (N/Tert-1-E6E7), E6 8S9A10T (N/Tert-1-E6 8S9A10T), E6 I128T (N/Tert-1-E6 I128T), and E7 ΔDLYC (N/Tert-1-E7 ΔDLYC) were previously generated [88,89] and maintained in keratinocyte serum-free medium supplemented with epidermal growth factor (EGF), bovine pituitary extract, and penicillin/streptomycin (Thermo Fisher). The mouse oropharyngeal epithelial (MOE) cell lines, NiMOE, mEERL, and MOE/shPtpn13, were obtained from John Lee [39] and cultured in E-medium (DMEM and F12 media supplemented with 0.005% hydrocortisone, 0.05% transferrin, 0.05% insulin, 0.0014% triiodothyronine, 0.005% EGF, and 2% FBS) as previously described [90].

### Flow cytometry

Single-cell suspensions were prepared, counted, and analyzed using specific antibodies (**S2 Table**) by an LSRII flow cytometer (BD Biosciences), as previously described [91]. Data were analyzed using the FlowJo software (Tree Star). Apoptotic cells were detected using the PE Annexin V Apoptosis Detection Kit with 7-AAD according to the manufacturer's protocol (BioLegend) and analyzed using an LSRII flow cytometer.

### Lentivirus production and transduction

The shRNAs targeting human *MARCHF8* were purchased from Sigma-Aldrich. The sgRNAs targeting mouse *Marchf8* were designed by the web-based software ChopChop

(http://chopchop.cbu.uib.no) [92]. The sgRNAs were synthesized and cloned into the lenti-CRISPR v2-blast plasmid (a gift from Mohan Babu, Addgene plasmid #83480) following BsmBI restriction enzyme digestion by ligating duplex oligonucleotides containing complementary overhangs purchased from Integrated DNA Technologies (IDT). All shRNA and sgRNA sequences are listed in **S3 and S4 Tables**, respectively. Lentiviruses containing shRNA or sgRNA were produced using 293FT cells with packaging constructs pCMV-VSVG and pCMV-Delta 8.2 (gifts from Jerome Schaack). The lentiviruses were collected 48 hrs post transfection and concentrated by ultracentrifugation at 25,000 rpm for 2 hrs. Cells were incubated with lentiviruses for 48 hrs in the presence of polybrene (8 μg/ml) and selected with puromycin (2 μg/ml) or blasticidin (8 μg/ml), respectively.

## DNA-protein pulldown assay

The assay was performed as previously described [93]. Briefly, biotinylated 90 bp oligonucleotides containing the *MARCHF8* promoter sequence between -85 and +5 (wildtype or the E-box mutants) (**Figs 2C and S3A**) were synthesized and incubated with 100 μl of M-280 streptavidin-coated magnetic beads (Dynal) for 1 h. The beads were collected using a magnetic bead concentrator (Dynal), washed with 1X binding and washing (B&W) buffer (10 mM Tris-HCl, pH 7.5, 1 mM EDTA, 1 M NaCl) and TEN buffer (10 mM Tris-HCl, pH 7.5, 1 mM EDTA, 0.1 M NaCl), and incubated with 500 μg of cell nuclear extracts for 2 hrs at 4°C in the presence of 3 μg of poly (dI-dC). After washing with TEN buffer, bound proteins were eluted using 100 μl of 1X B&W buffer and analyzed by western blotting and mass spectrometry.

## Mice and tumor growth

C57BL/6J mice were obtained from Jackson Laboratory and maintained following the USDA guidelines. 6 to 8-week-old mice were injected with $5 \times 10^5$ mEERL cells subcutaneously into the rear right flank ($n$ = 10 per group). Tumor volume was measured twice a week and calculated using the equation: volume = (width$^2$ X length)/2. Animals were euthanized when tumor volume reached 2000 mm$^3$, as previously described [94]. Conversely, mice were considered tumor-free when no measurable tumor was detected for 12 weeks. Survival graphs were calculated by standardizing for a tumor volume of 2000 mm$^3$. The Michigan State University Institutional Animal Care and Use Committee (IACUC) approved experiments by the National Institutes of Health guidelines for using live animals.

## Quantitative reverse transcription-PCR (RT-qPCR)

Total RNA was isolated using RNeasy Plus Mini Kit (Qiagen). First-strand cDNA was synthesized from 2 μg of total RNA using reverse transcriptase (Roche). Quantitative PCR (qPCR) was performed in a 20 μl reaction mixture containing 10 μl of SYBR Green Master Mix (Applied Biosystems), 5 μl of 1 mM primers, and 100 ng of cDNA templates using a Bio-Rad CFT Connect thermocycler. Data were normalized to glyceraldehyde 3-phosphate dehydrogenase (GAPDH). Primers used in qPCR (**S5 Table**) were synthesized by IDT.

## Immunoprecipitation (IP) and western blotting

Whole-cell lysates were prepared in 1X radioimmunoprecipitation assay buffer (RIPA) buffer (Abcam) containing protease inhibitor cocktail (Roche) according to the manufacturer's instruction. Total protein concentrations were determined using the Pierce BCA Protein Assay Kit (Thermo Fisher). IP was performed using the Pierce Classic Magnetic IP/Co-IP Kit (Thermo Fisher). Briefly, 25 μl of protein A/G magnetic beads were incubated with 5 μg of

specific antibodies (**S2 Table**) for 2 hrs. 1 mg of the whole cell lysates were incubated with the antibodies coupled beads overnight at 4˚C. Western blotting was performed with 10–20 μg of total protein using antibodies listed in **S2 Table** as previously described [95]. Band densities were determined using ImageJ software and normalized to the β-actin band intensity.

### Promoter reporter constructs and luciferase assay

Luciferase reporter plasmids containing the *MARCHF8* promoter flanked by firefly luciferase were constructed with the backbone of the pGL4 basic vector (Promega). Six DNA fragments representing the *MARCHF8* promoter spanning around the transcription start site of the *MARCHF8* gene were cloned in the pGL4 basic vector using the HindIII and KpnI restriction sites. The 0.25 kb fragments of *MARCHF8* promoter mutated in either or both two E-boxes were generated using the QuikChange II Site-Directed Mutagenesis Kit (Agilent) using cloning primers (**S5 Table**). The luciferase assay was carried out as previously described [96]. Briefly, $2 \times 10^5$ cells were cotransfected with one of the pGL4-*MARCHF8* promoter constructs and pNCMV 16E6no* (Addgene plasmid #37454), CMV 16 E7 (Addgene plasmid #13686), pNCMV 16E6no* Y54D (Addgene plasmid #47705), pNCMV 16E6no* I128T (Addgene plasmid # 47706), pCMV 16E7 del DLYC (Addgene plasmid #13687) or the empty vector pGL4-Basic (Promega) along with pEF1α-RL as a normalization control (Promega). All E6 and E7 expression plasmids were gifts from Karl Munger. The cells were harvested after 48 hrs, and luciferase activity was measured using the Dual-Luciferase Reporter Assay System (Promega) according to the manufacturer's instructions.

### Liquid chromatography-mass spectrometry

Protein samples were mixed with 250 μL of 50 mM ammonium bicarbonate (pH 8.0) and incubated with Tris(2-carboxyethyl) phosphine and chloroacetamide at 10 mM and 40 mM, respectively, for 5 min at 45˚C with shaking at 2000 rpm. Trypsin in 50 mM ammonium bicarbonate was added to 10 ng, and the mixture was incubated at 37˚C overnight with shaking at 1500 rpm. The final volume of each digest was ~300 μL. After digestion, the samples were acidified to 1% Trifluoroacetic acid and subjected to C18 solid phase clean-up using StageTips to remove salts as described [97]. An injection of 5 μL was automatically made using a Thermo EASY-nLC injector onto a Thermo Acclaim PepMap RSLC 0.075 mm x 150 mm C18 column and washed for ~5 min using buffer A. Bound peptides were eluted with a gradient of 5% B to 25% B from 0 min to 19 min, 25% to 90% from 19 min to 24 min and held at 90% B for the duration of the run (Buffer A = 99.9% Water/0.1% Formic Acid, Buffer B = 80% Acetonitrile/ 0.1% Formic Acid/19.9% Water) at a constant flow rate of 300 nL/min. The column temperature was maintained at a constant temperature of 50˚C using an integrated column oven (PRSO-V1, Sonation GmbH). Eluted peptides were sprayed into a ThermoScientific Q-Exactive mass spectrometer using a FlexSpray spray ion source. Survey scans were taken in the Orbitrap (35,000 resolution, determined at m/z 200). The top 15 ions in each survey scan are then subjected to automatic higher energy collision-induced dissociation (HCD) with fragment spectra acquired at 17,500 resolutions. The resulting MS/MS spectra are converted to peak lists Mascot Distiller and searched against a database containing all human protein sequences available from Uniprot appended with common laboratory contaminants (downloaded from the cRAP project) using the Mascot searching algorithm. The Mascot output was then analyzed using a Scaffold to probabilistically validate protein identifications. Assignments validated using the Scaffold 1% FDR confidence filter are considered true. The exclusive hits were selected based on positive interaction with the MARCHF8 promoter DNA fragments but not with the negative control in both replicates at least 3-fold or higher.

## Statistical analysis

Data were analyzed using GraphPad Prism and were presented as mean ± standard deviation. Statistical significance was determined using an unpaired Student's *t-test*. $P$ values <0.05 are considered statistically significant. Distributions time-to-event outcomes (e.g., survival time) were summarized with Kaplan–Meier curves and compared across groups using the log-rank test with α = 0.01. Data are deposited in the Dryad Data Repository: https://doi.org/10.5061/dryad.ffbg79d04 [98].

## Supporting information

**S1 Fig. Protein and mRNA expression levels of HPV16 E6 and E7 in N/Tert-1 cells.** The HPV16 E7 protein levels were determined in N/Tert-1 cells and N/Tert-1 cells expressing HPV16 E6, E7, or E6 and E7 (E6E7) using western blotting (**A**). β-actin was used as a loading control. The size of HPV16 E7 in N/Tert-1 E7 cells is about 22 kDa because the protein is fused to the HA tag, while the size of HPV16 E7 in N/Tert-1 E6E7 cells is about 17 kDa because the protein is untagged. Total RNA was extracted from N/Tert-1 containing an empty vector and N/Tert-1 cells expressing HPV16 E6, E7, or E6 and E7 (E6E7). The HPV16 E6 (**B**) and E7 (**C**) mRNA expression levels were quantified by RT-qPCR. The data shown are normalized by the GAPDH mRNA level as an internal control. All experiments were repeated at least three times, and the data shown are means ± SD.
(TIFF)

**S2 Fig. *MARCHF8* expression is independent of p53 and E6AP and pRb binding domains of HPV16 E6 and E7, respectively.** MARCHF8 mRNA (**A**) and protein (**B—D**) were determined in N/Tert-1 cells and N/Tert-1 cells expressing HPV16 E6, E6 8S9A10T, and E6 I128T (**A** and **B**) or E7 and E7 ΔDLYC (**A** and **C**) using RT-qPCR and western blotting, respectively. RT-qPCR was performed using total RNA extracted from N/Tert-1 cells, and the data shown are normalized by the GAPDH mRNA level as an internal control (**A**). Western blotting of E6 and p53 (**B**) or E7 and pRb (**C**) was performed using N/Tert-1 cell lysates with β-actin as a loading control. The relative band density was quantified using Image (**D**). All experiments were repeated at least three times, and the data is shown as mean ± SD.
(TIFF)

**S3 Fig. The HPV oncoprotein E6 induces the *MARCHF8* promoter activity independent of the p53 and E6AP binding domains.** The map of the extended *MARCHF8* promoter region (-1340 to +160) is shown with the positions of two E-boxes (red) and transcription start sequence (TSS, gray) (**A**). The promoter-reporter construct (-1340 to +160) was transfected into HPV- (SCC1) cells and cotransfected with plasmids expressing wildtype E6, E6 Y54D, E6 I128T deficient in E6AP binding, wildtype E7, or E7 ΔDLYC deficient in pRb binding (**B**). Luciferase activity was measured 48 h post transfection. Representative data from three independent experiments are shown as a fold change relative to the empty pGL4.2 vector (Basic). $P$ values were determined by Student's *t*-test. $^*p < 0.05$, $^{**}p < 0.01$, $^{***}p < 0.001$.
(TIFF)

**S4 Fig. mRNA expression levels of FAS, TRAIL-R1, and TRAIL-R2 in HPV+ and HPV-cells.** The FAS, TRAIL-R1, and TRAIL-R2 mRNA expression levels in normal (N/Tert-1), HPV+ HNC (SCC2, SCC90, and SCC152), and HPV- HNC (SCC1, SCC9, and SCC19) cells (**A-C**) and N/Tert-1 cells expressing HPV16 E6, E7, or E6 and E7 (**D-F**) were quantified by RT-qPCR. The data shown are normalized by the GAPDH mRNA level as an internal control. All experiments were repeated at least three times, and the data shown are means ± SD. $P$

values were determined by Student's *t*-test. $^*p < 0.05$, $^{**}p < 0.01$, $^{***}p < 0.001$.
(TIFF)

**S5 Fig. Knockdown of *MARCHF8* expression increases FAS, TRAIL-R1, and TRAIL-R2 protein expression in HPV+ HNC cells.** HPV+ HNC (SCC2) cells were transduced with three lentiviral shRNAs against MARCHF8 (shR-MARCHF8 clones 3–5) along with scrambled shRNA (shR-scr). Protein expression of MARCHF8, FAS, TRAIL-R1, and TRAIL-R2 was determined by western blotting (**A**). Relative band density was quantified using NIH ImageJ (**B**). β-actin was used as an internal control. The data shown are means ± SD of three independent experiments. Cell surface expression of FAS (**C** and **D**), TRAIL-R1 (**E** and **F**), and TRAIL-R2 (**G** and **H**) proteins were analyzed by flow cytometry. Mean fluorescence intensities (MFI) of three independent experiments are shown (**D**, **F**, and **H**). *P* values were determined by Student's *t*-test. $^*p < 0.05$, $^{**}p < 0.01$, $^{***}p < 0.001$.
(TIFF)

**S6 Fig. mRNA expression levels of FAS, TRAIL-R1, and TRAIL-R2 by knockdown of *MARCHF8* expression.** Two HPV+ HNC cell lines, SCC152 (**A-C**) and SCC2 (**D-F**) were transduced with five and three lentiviral shRNAs against MARCHF8 (shR-MARCHF8), respectively, or scrambled shRNA (shR-scr). The mRNA levels of FAS (**A** and **D**), TRAIL-R1 (**B** and **E**), and TRAIL-R2 (**C** and **F**) were quantified by RT-qPCR. The data shown are normalized by the GAPDH mRNA level as an internal control. All experiments were repeated at least three times, and the data shown are means ± SD. *P* values were determined by Student's *t*-test. $^{***}p < 0.001$.
(TIFF)

**S7 Fig. MARCHF8 protein ubiquitinates FAS, TRAIL-R1, and TRAIL-R2 proteins.** Ubiquitinated proteins were pulled down from the cell lysate of HPV+ HNC (SCC152) cells with scrambled shRNA (shR-scr) or shRNA against MARCHF8 (shR-MARCHF8 clone 3) treated with a proteasome inhibitor MG132 using an anti-ubiquitin antibody (**A—F**). FAS (**A and B**), TRAIL-R1 (**C and D**), and TRAIL-R2 (**E and F**) proteins were detected in the immunoprecipitated proteins by western blotting.
(TIFF)

**S1 Table. MARCHF8 promoter binding proteins.**
(PDF)

**S2 Table. List of the antibodies.**
(PDF)

**S3 Table. List of the shRNAs.**
(PDF)

**S4 Table. List of the sgRNAs.**
(PDF)

**S5 Table. List of the oligonucleotides.**
(PDF)

## Acknowledgments

We thank Bardees Foda and members of the Pyeon laboratory for their valuable comments and suggestions. We thank Douglas Whitten (Mass Spectrometry and Metabolomics Core) and Daniel Vocelle (Flow Cytometry Core) at Michigan State University for their assistance

with mass spectrometry and flow cytometry analyses, respectively. We also thank Karl Munger, Jerome, Schaack, John Lee, and Mohan Babu for their gifts of plasmid constructs.

## Author Contributions

**Conceptualization:** Mohamed I. Khalil, Dohun Pyeon.

**Data curation:** Mohamed I. Khalil, Dohun Pyeon.

**Formal analysis:** Mohamed I. Khalil, Dohun Pyeon.

**Funding acquisition:** William C. Spanos, Dohun Pyeon.

**Investigation:** Mohamed I. Khalil, Canchai Yang, Lexi Vu, Smriti Chadha, Harrison Nabors, Dohun Pyeon.

**Methodology:** Mohamed I. Khalil, Canchai Yang, Lexi Vu, Dohun Pyeon.

**Project administration:** Mohamed I. Khalil, Dohun Pyeon.

**Resources:** Mohamed I. Khalil, Craig Welbon, Claire D. James, Iain M. Morgan, William C. Spanos.

**Supervision:** Mohamed I. Khalil, Iain M. Morgan, Dohun Pyeon.

**Validation:** Mohamed I. Khalil, Canchai Yang, Lexi Vu, Smriti Chadha, Harrison Nabors, Dohun Pyeon.

**Visualization:** Mohamed I. Khalil, Dohun Pyeon.

**Writing – original draft:** Mohamed I. Khalil, Dohun Pyeon.

**Writing – review & editing:** Mohamed I. Khalil, Lexi Vu, Craig Welbon, Claire D. James, Iain M. Morgan, Dohun Pyeon.

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
