## [Decision Letter · Decision Letter 0]

10 Nov 2022

Dear Dr. Pyeon,

Thank you very much for submitting your manuscript "HPV E6 upregulates MARCHF8 ubiquitin ligase and inhibits apoptosis by degrading the death receptors in head and neck cancer" for consideration at PLOS Pathogens. As with all papers reviewed by the journal, your manuscript was reviewed by members of the editorial board and by several independent reviewers. In light of the reviews (below this email), we would like to invite the resubmission of a significantly-revised version that takes into account the reviewers' comments.

The reviewers felt that the study is well done, clearly written and will contribute importantly to understanding virus/host interactions. However they had some concerns that need to be addressed.

First, contrary to the title, the data show that both E6 and E7 can increase levels of MarchF8. This needs to be addressed and mutants of E6 should be use to better address the mechanism.

Second, one reviewer points out that the difference in expression of MARCHF8 comparing HPV + versus HPV- in head and neck tumors is relatively small and could be explained by a number of other factors. These should be included in interpreting the results.

Third, Multiple replicates of the MARCHF8/Fas/Ub (and other death receptor) complex pulldowns in Fig 5 should be quantified.

We cannot make any decision about publication until we have seen the revised manuscript and your response to the reviewers' comments. Your revised manuscript is also likely to be sent to reviewers for further evaluation.

Sincerely,

Denise A. Galloway

Academic Editor

PLOS Pathogens

Alison McBride

Section Editor

PLOS Pathogens

Kasturi Haldar

Editor-in-Chief

PLOS Pathogens

orcid.org/0000-0001-5065-158X

Michael Malim

Editor-in-Chief

PLOS Pathogens

orcid.org/0000-0002-7699-2064

The reviewers felt that the study is well done, clearly written and will contribute importantly to understanding virus/host interactions. However they had some concerns that need to be addressed.

First, contrary to the title, the data show that both E6 and E7 can increase levels of MarchF8. This needs to be addressed and mutants of E6 should be use to better address the mechanism.

Second, one reviewer points out that the difference in expression of MARCHF8 comparing HPV + versus HPV- in head and neck tumors is relatively small and could be explained by a number of other factors. These should be included in interpreting the results.

Third, Multiple replicates of the MARCHF8/Fas/Ub (and other death receptor) complex pulldowns in Fig 5 should be quantified.

Reviewer's Responses to Questions

**Part I - Summary**

Reviewer #1: This work by Khalil et al . investigates the modulation of the cellular E3 Ub ligase MARCHF8 by human papillomaviruses. MARCHF8 downregulates surface levels of immune receptors including CD86, MHC-II, and TNF superfamily death receptors via Ub-dependent mechanisms, and HPV+ cells show decreased surface levels of many of these receptors. Using their previously published transcriptomics data from clinical samples, they show statistically significant upregulation of MARCHF8 in HPV+ but not HPV- head and neck cancers compared to healthy controls. Similar trends were seen in HPV+ versus HPV- head and neck SCC lines. They use promoter/reporter assays and find that E6 drives increased MARCHF8 expression by activation of Myc/Max which bind to E-boxes in the MARCHF8 promoter. Silencing of MARCHF8 restored surface levels of death receptors and led to enhanced apoptosis in response to ligands. KO of MARCHF8 in mouse cancer cells expressing E6/E7 caused increased apoptosis and decreased tumor growth in vivo. Overall conclusions are that HPV commandeers MARCHF8 to decrease surface expression of death receptors, thereby evading death receptor mediated apoptosis. These findings are highly significant for both the HPV life cycle (virology) and HPV-associated cancers.

Overall this significant work is of high rigor. Multiple systems/models were used and appropriate controls were done. The data strongly supports their conclusion. Only one recommendation is outlined below.

Reviewer #2: Khalil….Pyeon et al.

“HPV E6 upregulates MARCHF8 ubiquitin ligase and inhibits apoptosis by degrading

the death receptors in head and neck cancer.”

PPATHOGENS-D-22-0169.

Herpesviruses express mutant analogues of the host-cell ubiquitin ligase MARCH8 that has been attributed as a negative regulator of cellular death-receptors. Inspired by these observations, the authors examined if MARCH8 might be involved in papillomavirus pathogenesis in head and neck cancers. This study is thus a hypothesis directed examination designed to elicit evidence that papillomavirus oncogenes confer traits of altered death receptor function via manipulation of MARCH8. While this is a potentially powerful approach, it is also subject to confirmation bias.

The authors have examined previously acquired RNA expression data derived from microarray hybridizations for the expression of MARCHF8 ubiquitin ligase expression in head and neck cancers. Expression of MARCH8 was compared between multiple cancer samples that were either HPV+ or HPV- or to multiple samples of normal epithelium (site not otherwise specified). They observed modestly elevated expression of MARCH8 RNA in the HPV+ compared to the normal epithelium (Fig 1A). Although the large number of samples allows for statistical significance, it is likely that most of the normal epithelial sample is comprised of differentiated cells that may have lower MARCH8 expression compared to proliferative basal cells---protein expression data for MARCH8 at the Protein Atlas is consistent with this in squamous epithelial samples (although no expression of oral mucosa was present there). So it is unclear to me if the modest expression data difference between normal and HPV+ cancers is meaningful even if statistical significance is reached. There was no statistical difference between either the normal to HPV- sample sets or between the HPV+ and HPV- cancer sets. So Fig. 1A analysis is not strongly supportive. In Fig. 1B, similar RNA abundance analysis is carried out between HPV+, HPV- cancer cell lines, and tert-immortalized epithelial cells showing larger differences between tert-immortalized epithelium and HPV+ tumor cell lines and more modest differences between the tert and HPV- cell lines. But at the protein expression level, the differences between the tert-immortalized cells and the same cells expressing either E6, E7, or E6+E7 are modest, being about 2-2.5 fold elevated compared to vector, and although all are statistically elevated compared to vector-transduced cells, the E6, E7, and E6/E7 samples are quite similar to each other. This undermines the notion that the trait is conferred by E6 (vs E7) and undermines the title and subsequent experimental approaches of the study. Another way to demonstrate specificity would be the use of mutants of E6 and E7 which is not done. A mutant that blocks effects one trait (such as altered MARCH expression) without changing another function (such as p53 or NHERF1 or PDZ expression in E6, or RB or PTPN14 degradation for E7) could be used to demonstrate specificity of action by E6 or E7 while also providing mechanistic insight. The comparison between HPV+ and HPV- cancers for MARCH8 expression is less compelling because we do not know if these two types of cancers even originate from the same epithelial cell type, meaning that other reasons, not contemplated by the authors, could account for the difference in MARCH8 expression between HPV+ and HPV- cancer cells.

Therefore, if the authors propose that expression levels of MARCH8 are responsible for survival traits in E6/E7 expressing cells, then a careful dose response relationship must be established between MARCH8 expression levels and cell death traits in the range of expression alterations observed to be conferred by E6/E7. This is the key relationship. There should also be a demonstration of specificity of effect using intelligently selected mutants of E6 and or E7.

Fig.2 demonstrates an effect between E-boxes and E6 induction of a synthetic promoter construct. Yet the effect is not compelling in that upon mutation of both E-boxes in the reporter plasmid, E6 activates the promoter 20X instead of 40X. That may be significant, but is it important? Given that they show E7 activates MARCH8 at the promoter level as well as E6 in TERT cells, I do not think Fig. 2 contributes to the arguments they are making.

In Fig. 3, the SCC2 cell line seem non-supportive of their hypothesis for a role of E6 and E7 in that while MARCH8 levels are high (from Fig. 1) TRAILR1 levels are not reduced. The more informative experiment is the transduction of tert immortalized cells with viral oncoproteins and FACS analysis of death receptor expression. In this case, E6, E7, and E6/E7 all reduce surface expression. E6 does not reduce levels seen in cells also transduced by E7. This is inconsistent with their hypothesis of specificity of action of E6 vs E7 and supports a non-specific mechanism at work in oncogene expressing tert cells. Again, a careful mutant analysis might support a specific mechanism that is not obvious as yet.

In Fig 4 the authors set up a crucial experiment, using shRNA to knock down MARCHF8 expression. It is a bit odd that all 5 shRNA knock down MARCHF8 to a similar degree, which has almost never been my experience with a set of 5 shRNA lentiviruses. This raises the possibility that the expression difference between scramble and the 5 shrna to MARCH8 is a difference that could be attributed to the scramble control. Expression of an shRNA escape MARCHF8 mRNA might resolve this possibility. The data does show that compared to scramble all 5 shRNA reduce MARCHF8 and increase death receptor expression as noted by the authors.

In Fig. 5c, which band is Ub-Fas? Overall, Fig. 5 is supportive of MARCHF8 having a role in the ubiquitination of Fas, Trail’s and thereby negative regulation. This corroborates earlier results from others.

Fig 6 supports the conclusion that when Marchf8 expression is manipulated by shRNA, apoptotic traits are altered as expected.

Fig 7 utilizes a mouse expression system comparing “normal” mouse oral epithelium, the same cells transformed with HPV and the same cells transformed with HPV. Here the data is less convincing. Starting cells are compared to HPV-transformed (not specified if retrovirus, plasmid, HPV genomes or what) and HPV-non-transformed cells (oncogenes used in the transformation unspecified). So we have no idea what mechanisms might be involved. But what is clear is that both types of transformation events induced the expression of marchf8, with fas and trail-r1 alteration in the expected pattern. However, and unexpectedly, Trail-2 was not reduced in HPV- transformants but decreased in HPV transformants.

In Fig. 8, marchf8 is knocked out using crispr and marchf8, fas, trail-r1 and trail-r2 expression follows the expected pattern. In Fig. 9 knockout of marchf8 reduces tumor growth rate in HPV expressing cancer cells, as one might expect from purported effects upon apoptosis, but also possibly from march8 effects upon other as yet undescribed effects upon other cellular transmembrane proteins.

In summary, there is much to like about this study. The connections between marchf8 expression of the expression of TNF and fas receptors looks pretty solid in several experimental systems, but that work is not clearly novel to this study. The connections between the expression of E6 and the regulation of marchf8 seem poorly worked out to me. First, it looks clear that both E6 and E7 change the expression of marchf8 and this is ignored. The connection between E6 and myc and marchf8 looks tenuous. There is no E6 mutant analysis to support the specificity of E6 action, and one could make a strong case that this is a fairly non-specific induction of marchf8 related to transformation in the particular cells utilized.

Reviewer #3: Overall, this well-conceived and well-executed work provides mechanistic details regarding how the HPV 16 E6 oncoprotein inhibits apoptosis in the virus’ host cells. As such, it advances the fields of HPV virology, virus/host interaction, the ubiquitin/proteasome pathway and cancer, and is anticipated to be a valuable addition to the literature.

**Part II – Major Issues: Key Experiments Required for Acceptance**

Reviewer #1: Multiple replicates of the MARCHF8/Fas/Ub (and other death receptor) complex pulldowns in Fig 5 should be quantified.

Comment: Its puzzling that E7 causes increased steady state levels of MARCHF8 transcript but has no effect on MARCHF8 transcription in the promoter assays. Is E7 known to modulate RNA stability of other cellular transcripts? Could their be additional E7-responsive elements further than 1kb upstream of the MARCHF8 ORF? Might E7 have an effect if the promoter assays are done in primary keratinocytes?

Reviewer #2: see above section

Reviewer #3: (No Response)

**Part III – Minor Issues: Editorial and Data Presentation Modifications**

Reviewer #1: (No Response)

Reviewer #2: The paper is well written and proof-read.

Reviewer #3: • The manuscript demonstrates that following E6 expression, FAS, TRAIL-R1 and TRAIL-R2 are targets of MARCHF8 for ubiquitinylation and degradation. Are these proteins considered to be natural or canonical targets of MARCHF8, or is this enhanced turnover a specific consequence of the virus?

• The authors state (line 51) that HPV is associated with about 25% of head and neck cancers, with no reference given. Estimates vary, but most recent estimates put the percentage of HN cancers that are HPV+ to be significantly higher than 25%.

• The authors note that both E6 and E7 can upregulate expression of MARCHF8, but found that when using the plasmid reporter system, only E6 gave a positive signal. They then note that there may be an alternative mechanism associated with E7. It would have been helpful, perhaps in the discussion, to speculate on what some possible mechanisms might be.

• When assessing apoptosis, the authors used FasL and anti-Fas to induce the process. It would have been interesting to see the impact of modulating MARCHF8 expression on apoptosis induced through other receptors (like TRAIL or TNF), or even through a pathway that routes through p53. This would have helped to further define the scope of this particular MARCHF8-dependent mechanism.

• In the mouse oral cancer cells, while there is data provided on surface expression of Fas and TRAIL-R2, no data is given for TRAIL-R1. The total level of TRAIL-R1 is addressed.

• The authors note the death domain carried by the target proteins. What other proteins carry this same domain, and is it anticipated that some or all of them might also be targets for MARCHF8?

• Figure 10 is a helpful schematic. However, it is not referred to in the text.

PLOS authors have the option to publish the peer review history of their article (what does this mean?). If published, this will include your full peer review and any attached files.

Reviewer #1: No

Reviewer #2: No

Reviewer #3: No
---

## [Editor Report · Decision Letter 1]

1 Feb 2023

Dear Dr. Pyeon,

We are pleased to inform you that your manuscript 'HPV upregulates MARCHF8 ubiquitin ligase and inhibits apoptosis by degrading the death receptors in head and neck cancer' has been provisionally accepted for publication in PLOS Pathogens.

Best regards,

Denise A. Galloway

Academic Editor

PLOS Pathogens

Alison McBride

Section Editor

PLOS Pathogens

Kasturi Haldar

Editor-in-Chief

PLOS Pathogens

orcid.org/0000-0001-5065-158X

Michael Malim

Editor-in-Chief

PLOS Pathogens

orcid.org/0000-0002-7699-2064

The authors have answered the reviewers concerns successfully.
---

## [Editor Report · Acceptance letter]

28 Feb 2023

Dear Dr. Pyeon,

We are delighted to inform you that your manuscript, "HPV upregulates MARCHF8 ubiquitin ligase and inhibits apoptosis by degrading the death receptors in head and neck cancer," has been formally accepted for publication in PLOS Pathogens.

Best regards,

Kasturi Haldar

Editor-in-Chief

PLOS Pathogens

orcid.org/0000-0001-5065-158X

Michael Malim

Editor-in-Chief

PLOS Pathogens

orcid.org/0000-0002-7699-2064